# NoVo: Norm Voting off Hallucinations with Attention Heads in Large Language Models

**Zhengyi Ho[1], Siyuan Liang[1*], Sen Zhang[2], Yibing Zhan[2], Dacheng Tao[1*]**
{zhengyi001, siyuan.liang, dacheng.tao}@ntu.edu.sg
{senzhang.thu10, zhanybjy}@gmail.com
[1]Nanyang Technological University, Singapore 639798
[2]University of Sydney  *Corresponding Author

## ABSTRACT

Hallucinations in Large Language Models (LLMs) remain a major obstacle, particularly in high-stakes applications where factual accuracy is critical. While representation editing and reading methods have made strides in reducing hallucinations, their heavy reliance on specialised tools and training on in-domain samples, makes them difficult to scale and prone to overfitting. This limits their accuracy gains and generalizability to diverse datasets. This paper presents a lightweight method, Norm Voting (NoVo), which harnesses the untapped potential of attention head norms to dramatically enhance factual accuracy in zero-shot multiple-choice questions (MCQs). NoVo begins by automatically selecting truth-correlated head norms with an efficient, inference-only algorithm using only 30 random samples, allowing NoVo to effortlessly scale to diverse datasets. Afterwards, selected head norms are employed in a simple voting algorithm, which yields significant gains in prediction accuracy. On TruthfulQA MC1, NoVo surpasses the current state-of-the-art and all previous methods by an astounding margin—at least 19 accuracy points. NoVo demonstrates exceptional generalization to 20 diverse datasets, with significant gains in over 90% of them, far exceeding all current representation editing and reading methods. NoVo also reveals promising gains to finetuning strategies and building textual adversarial defence. NoVo's effectiveness with head norms opens new frontiers in LLM interpretability, robustness and reliability. Our code is available at: https://github.com/hozhengyi/novo

## 1 INTRODUCTION

One of the most significant challenges facing Large Language Models (LLMs) is their tendency to hallucinate—outputs that are factually incorrect or entirely fabricated (Zhang et al., 2023b). This flaw is particularly serious in high-stakes applications like finance and healthcare, where even small errors can lead to huge losses and compromised patient safety (Kang & Liu, 2023; Pal et al., 2023a). Reducing factual hallucinations is a critical research area with major practical benefits, essential for realising the full potential of LLMs to revolutionise these industries by enhancing efficiency and decision-making, and safeguarding against costly and harmful errors (Kaddour et al., 2023).

Given these serious risks and the high cost of retraining LLMs, it is crucial to find affordable techniques to reduce factual hallucinations. Although inference techniques such as retrieval augmentation and prompt engineering work well, they come with significant limitations: latency and external dependencies, and the need for user expertise, respectively (Zhao et al., 2024; Sahoo et al., 2024). In response, we turn to representation editing and reading methods (REAR) (Zou et al., 2023), which operate within the model, ensuring rapid response times and eliminating the need for external data or user interaction. REAR methods reduce hallucinations by modifying or extracting factual information encoded in LLMs' latent feature vectors (hidden states), such as attention heads (Bronzini et al., 2024). This process often requires specialised tools such as probes and autoencoders (Li et al., 2024; Zhang et al., 2024), trained and tuned on in-domain samples. Thus, existing REAR methods are difficult to scale and prone to overfitting, leading to limited accuracy gains and generalizability to diverse datasets. Tackling these limitations is crucial, since REAR methods can improve factuality with minimal costs, latency, and user friction; highly desirable attributes for practical applications.

This paper presents **No**rm **Vo**ting (NoVo), a more accurate REAR method for reducing factual hallucinations in diverse multi-choice scenarios. NoVo works by efficiently measuring *latent truth* (Zou et al., 2023) in certain attention head norms, thus avoiding the log likelihood layer, which can induce hallucinations by favouring fluency over factuality (Ji et al., 2023). NoVo first selects attention head ***norms*** that correlate with truth using only inference on 30 random samples, allowing NoVo to scale to numerous datasets. Then, selected head norms participate in majority ***voting*** as an ensemble of weak learners (Schapire, 1990) to boost accuracy. The process is summarised in Figure 1. NoVo is made lightweight by design for scalable use, requiring no specialised tools or training, which enables wide evaluations across a diverse range of reasoning, factuality, and understanding datasets. To our knowledge, we are the first to explore attention head norms as a measure of latent truth. This raises exciting questions about their wider roles in interpreting and addressing hallucinations.

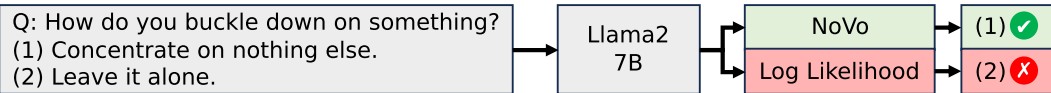

Figure 1: **Overview of our method**. NoVo improves factuality over the log likelihood.

On TruthfulQA MC1, an unsolved hallucination benchmark, NoVo achieves a new state-of-the-art (SOTA) accuracy of 78.09% on a 7B model, substantially outperforming the log likelihood and the best REAR method by at least 24 and 19 accuracy points. NoVo scales and generalizes well to 20 diverse datasets featuring varied topics and formats, with significant gains in over 90% of them, dramatically surpassing previous REAR methods which were only evaluated on a few factuality benchmarks. Additionally, NoVo achieves promising gains on AdversarialGLUE and DeBERTa finetuning. We choose to evaluate NoVo on multiple-choice questions (MCQs) because they test foundational cognitive skills (Bloom et al., 1964), are critical for real-life applications in high-stakes standardised assessments (OSHA, 2016; Pal et al., 2023b; NCBE, 2024; ETS, 2024), offer an objective metric across diverse benchmarks for academic evaluation (Mee et al., 2024), and pose unique technical challenges that reveal core limitations and internal misalignments within LLMs (Zheng et al., 2024; Kiela et al., 2021). We analyse why head norms are correlated with truthfulness, and find that they reliably spike by up to 83% in token positions of both factual proposition completions and pertinent factual associations despite misleading contexts. Beyond MCQs, our findings reveal a more fundamental, task-agnostic problem in high-stakes scenarios: factual hallucinations caused by misalignments between hidden states and the language likelihood. NoVo's strong performance across diverse and representative benchmarks demonstrates that this problem can be mitigated using head norms. These norms show promise in reliably ranking truth across multiple candidate spans during decoding or retrieval, and in future works for enhancing model robustness and alignment during fine-tuning, ultimately contributing to more trustworthy generations in high-stakes scenarios.

Our main contributions can be summarised in three points: ❶ We use head norms to accurately rank truthfulness between several candidate texts, evaluated on diverse MCQs. ❷ We show and explain the correlation between head norms and truth. ❸ We demonstrate and mitigate a fundamental cause of factual hallucinations: language likelihood misalignments with internal states, using TruthfulQA.

## 2  RELATED WORKS

**Representation Editing**   Some REAR methods involve manually modifying hidden states during inference towards hidden state clusters, formed by the forward pass of true and false sequences (Burns et al., 2023), as a generic hallucination mitigation technique. All methods here require cross-fold training on in-domain samples from the test set, with some set aside for validation. Inference Time Intervention (ITI) edits specific attention head hidden states towards those clusters (Li et al., 2024), using custom-built linear probes and visualisation tools. Similarly, TruthForest (TrFr) edits heads toward multiple directions (Chen et al., 2024), while Truthx edits concepts of truth disentangled from hidden states with a deep autoenconder (Zhang et al., 2024), as a specialised tool.

**Representation Reading**   There are decoding strategies that use hidden states to improve the factuality of LLMs without editing. Decoding by Contrasting Layers (DoLa) (Chuang et al., 2024), tuned with in-domain samples, extracts factual information in intermediate layers. Induce-then-Contrast Decoding (ICD) (Zhang et al., 2023a) contrasts LLM outputs with a special hallucinatory model

trained on an external dataset. RePE (Zou et al., 2023) relies on curated templates and samples to measure truth in hidden states using a specialised tool known as linear artificial tomography. All these methods, except RePE, are generic hallucination mitigation techniques without editing.

Unlike current REAR methods, NoVo uses only attention head norms and does not require any external modules, custom-built probes, special techniques, in-domain sample training, or curated resources. This makes NoVo lightweight, enabling it to scale and generalize to numerous MCQ tasks. Together with a simple voting algorithm, NoVo is also significantly more accurate.

## 3 METHOD

### 3.1 BACKGROUND

**Prior Insights**  Previous studies have demonstrated that hidden states in the multi-layer perceptron (MLP) modules of LLMs can be linearly classified into true-false clusters (Burns et al., 2023; Azaria & Mitchell, 2023; Zou et al., 2023). Further studies extended this idea to individual heads in the multi-head attention (MHA) module (Li et al., 2024; Chen et al., 2024). In computer vision, studies have shown that the L2 norm of the final feature vector in convolutional networks correlates with image quality (Kim & Lee, 2017; Yan et al., 2019). Insights from these works show that it is reasonable to expect the L2 norm of some heads, denoted $T$, to correlate with truth.

**Setup**  In the forward pass of an auto-regressive decoder transformer LLM, token sequences of length $s$ are embedded and featurized through multiple layers, each consisting of a MHA and MLP module, before reaching the logit layer for next-token prediction. An LLM with $L$ layers and $H$ heads per MHA will have a total of $LH$ heads throughout the network, excluding the logit and embedding layers. The MHA at layer $l \in \{1, 2 \dots, L\}$ takes as input $\boldsymbol{X}^{(l-1)} \in \mathbb{R}^{s \times d}$ from the previous layer and projects each feature in the sequence to their key, query and value states

$$\boldsymbol{Q}^l = \boldsymbol{X}^{(l-1)} \boldsymbol{W}^l_{query} \qquad \boldsymbol{K}^l = \boldsymbol{X}^{(l-1)} \boldsymbol{W}^l_{key} \qquad \boldsymbol{V}^l = \boldsymbol{X}^{(l-1)} \boldsymbol{W}^l_{value} \qquad (1)$$

ignoring the bias term, where $\boldsymbol{Q}^l, \boldsymbol{K}^l, \boldsymbol{V}^l \in \mathbb{R}^{s \times d}$ and $d$ is the model dimension. Splitting them on the column axis gives $\boldsymbol{Q}^{l,h}, \boldsymbol{K}^{l,h}, \boldsymbol{V}^{l,h} \in \mathbb{R}^{s \times d'}$ for $h \in \{1, 2 \dots, H\}$ and $d' = d/H$. The context vectors, or attention heads, $\boldsymbol{C}^{l,h} \in \mathbb{R}^{s \times d'}$, are thus computed via the attention mechanism as

$$\boldsymbol{C}^{l,h} = \boldsymbol{A}^{l,h} \boldsymbol{V}^{l,h} \qquad\qquad \boldsymbol{A}^{l,h} = \text{softmax}\left( \frac{\boldsymbol{Q}^{l,h} (\boldsymbol{K}^{l,h})^T}{\sqrt{d'}} + \boldsymbol{M} \right), \qquad (2)$$

where $\boldsymbol{M}$ enforces auto-regression by setting $\boldsymbol{A}^{l,h}$ to a lower triangular matrix. In Equation 2, each head in the sequence $\boldsymbol{C}^{l,h}$ is the attention weighted sum of each value state in $\boldsymbol{V}^{l,h}$, computed component-wise from the current and all previous sequence positions as

$$\boldsymbol{C}^{l,h} = \boldsymbol{A}^{l,h} \boldsymbol{V}^{l,h} = \begin{bmatrix} a_{11}v_{11} & \cdots & a_{11}v_{1d'} \\ \sum_{j=1}^{2} a_{2j}v_{j1} & \cdots & \sum_{j=1}^{2} a_{2j}v_{jd'} \\ \vdots & \ddots & \vdots \\ \sum_{j=1}^{s} a_{sj}v_{j1} & \cdots & \sum_{j=1}^{s} a_{sj}v_{jd'} \end{bmatrix} \qquad (3)$$

**Motivation**  Insights from prior works suggests a reasonable expectation that $T$ correlates with the truthfulness of a text sequence. Furthermore, LLMs encode diverse language features in their hidden states, which often self-organise along meaningful dimensions (Mikolov, 2013). It is plausible that some dimensions reflect the alignment of truthful propositions with reality, where the coherence of certain concepts such as passengers and planes, might consistently express itself as L2 norm magnitude changes. The expectation that L2 norms can measure truth as a broad and continuous scalar, is well-aligned with findings made by Lin et al. (2022), which framed truth as a probability measure, and Li et al. (2024), which proposed that latent truth is expressed in multiple directions. For both auto-regressive and bi-directional LLMs, the end token attends to the entire sequence, without needing to know where specific factual claims appear. Therefore, we define $T$ as the attention head norm at the final sequence position such that $T^{l,h} = \left\| \boldsymbol{C}^{l,h}_{-1,:} \right\|_2$. This process is shown in Figure 2. We do not assume the correlation direction and allow for inverse relationships as well. In Appendix

F, we show that the latter approach is better. Since $T^{l,h}$ is unbounded, with correlation direction and $l, h$ being unspecified, it cannot be used yet. $T^{l,h}$ instead forms the basis for NoVo, which addresses these issues and operationalises $T^{l,h}$ to improve factual accuracy in MCQs.

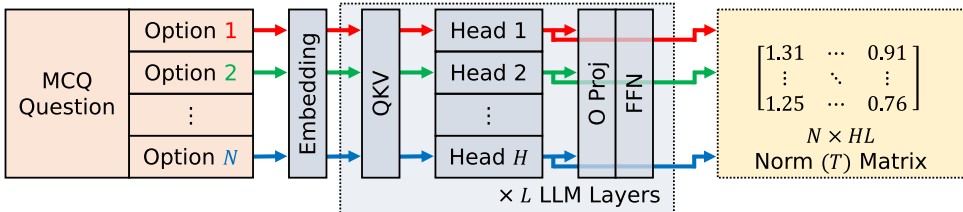

Figure 2: The **Norm Matrix** at the right contains all $T^{l,h}$ values taken throughout the LLM, but cannot be used to answer MCQs. Instead, this operation forms the basic building block of NoVo.

## 3.2 NORM VOTING (NOVO)

**Norm Selection**    The goal of this stage is to operationalise $T^{l,h}$ by resolving its unbounded nature, and specifying all $(l, h)$ indices that correlates with truth, including the correlation direction. Figure 3 shows this stage in five steps. In step ❶, 30 random samples are fed into the LLM to produce 30 Norm Matrices, packed as a tensor. The idea here is that all head norms are initially assumed to correlate with truth, each producing two predictions from the argmax and argmin operators. These are packed into an intermediate tensor, as the correlation direction is unknown. The unbounded nature of $T^{l,h}$ is resolved here, since both operators are relative. In step ❷, each head receives an accuracy score across 30 samples for both sets of prediction, forming a matrix with two rows representing each prediction set, and columns that represent each head's accuracy. It is clear here that most heads are poor performers. In steps ❸ and ❹, the correlation direction and strength are identified using these accuracies scores as a proxy measure. This approach does not require any training, special techniques or external tools, making NoVo lightweight and scalable. The row with the highest accuracy indicates the correlation direction. Steps ❹ and ❺ determines which heads are strongly correlated with truth, by taking the higher accuracy of the two rows. This is followed by a thresholding operation, set at the 85th percentile ($P_{85}$) of all accuracies. We refer to these remaining heads as "Voters". For clarity, $(l, h)$ is enumerated as consecutive integers, starting from 0 for the first head in the first layer. **This entire stage is only performed once**, as the Index Vector and Indicators are reused, and takes less than 10 seconds on one NVIDIA A100 GPU. The number of samples and threshold are hyper-parameters, found to be optimal at 30 and $P_{85}$. The search for these two values is detailed in Appendix B, with a hyper-parameter free variant explored in Appendix C.

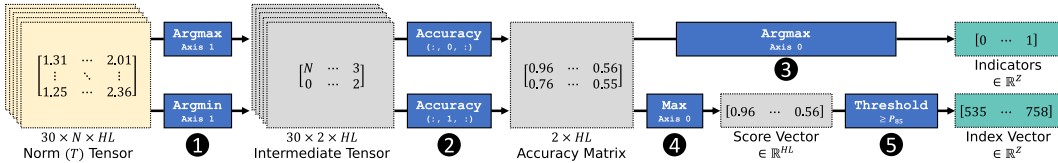

Figure 3: The selection stage uses the Norm Matrix from Figure 2 to determine the correlation direction of each $T^{l,h}$, serialised as **Indicators**. All $(l, h)$ indices that vary with truth are also specified in the **Index Vector**, expressed as enumerated integers for clarity.

**Voting Inference**    Now that the latent measure of truth $T^{l,h}$ is operationalised with NoVo, zero-shot MCQ classification can begin. The goal of this stage is to output more accurate predictions via majority voting, shown in four steps in Figure 4. In Step ❶, an example MCQ with three options is fed through the LLM to produce the Norm Matrix. Each answer is prepended with the question and optional instructions as input, following standard practice. In Step ❷, Voters are selected with the Index Vector from the previous stage. In Step ❸, the correlation direction of each Voter is flagged with Indicators, also from the previous stage. This allows for dynamic selection between the argmax or argmin operators, for individual Voter predictions. While each Voter's $T$ is unbounded and could become very large, we observe in practice that it is well-conditioned to varying truthfulness in a

sequence. In most cases, $T$ ranges between 0.5 to 3. In step ❹, all Voter predictions participate in a majority vote via the mode operator, resulting in the final MCQ prediction of the LLM.

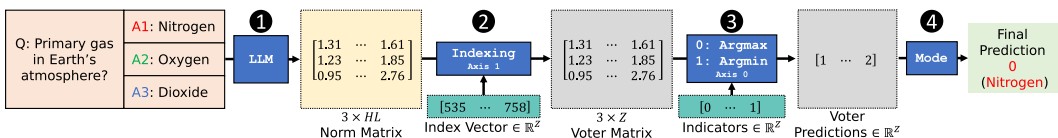

Figure 4: The voting stage uses the Norm Matrix from Figure 2, and the Indicators and Index Vector from Figure 3, to accurately answer MCQ questions during LLM inference.

## 4 EXPERIMENT AND DISCUSSION

### 4.1 SETTINGS

**Experiments**   We evaluate NoVo in three key areas: ❶ its effectiveness in reducing factual hallucinations compared to existing REAR methods, ❷ its generalizability across various reasoning and natural language understanding (NLU) tasks, and ❸ its adaptability to broader classification tasks, indicated by its finetuning performance. Experimental results for the first area are shown in Table 1. Results for the second area are presented in Tables 2, 3, and 4, while the third area is reported at the bottom of Table 2. To avoid over-reporting results, all experiments use 30 random training samples drawn without tuning for Norm Selection, and use zero-shot prompts without tuning (Perez et al., 2021). More information on experimental details, random variations, hidden state analysis, and additional models can be found in Appendices A, D, H, and I respectively.

**Models**   NoVo is evaluated in two classification settings: zero-shot and finetuned. Zero-shot is the primary setting used in most experiments, and the results are presented in Tables 1 through 4. Finetuning, on the other hand, is used in only one experiment, which is reported at the bottom of Table 2. In the zero-shot setting, NoVo is applied to four 7B decoder LLMs: Llama2 and LLama2-Chat (Touvron et al., 2023), Vicuna (Chiang et al., 2023) and Mistral-Instruct (Jiang et al., 2023). For the finetuned setting, NoVo is applied to DeBERTa-Large (He et al., 2023). Additionally, Table 2 includes results from two finetuned 11B models, UnifiedQA and UNICORN (Khashabi et al., 2020; Lourie et al., 2021), for reference purposes only, without making any direct comparisons.

**Datasets**   We evaluate NoVo's effectiveness in reducing factual hallucinations on TruthfulQA MC1 (Lin et al., 2022), a standard and unsolved hallucination benchmark used by all previous REAR methods. For our generalizability experiment, we apply NoVo to diverse datasets covering multiple topics and presented in various formats. This includes CommonsenseQA 2.0 (CQA2) (Talmor et al., 2021) for commonsense reasoning. QASC (Khot et al., 2020) tests for scientific knowledge. SWAG (Zellers et al., 2018) and HellaSwag (HSwag) (Zellers et al., 2019) requires sentence completions about challenging commonsense scenarios. SIQA (Sap et al., 2019) and PIQA (Bisk et al., 2020) looks for social and physical reasoning, respectively. CosmosQA (Cosmos) (Huang et al., 2019) requires causal reasoning over narrative contexts. CICERO V1 and V2 (CICv1, CICv2) (Ghosal et al., 2022b; Shen et al., 2022) tests for multi-turn dialogue and strategic reasoning. We use a MCQ variant from Ghosal et al. (2022a). Adversarial GLUE (AdvGLUE) (Wang et al., 2021) tests model robustness to adversarial texts in NLU tasks. FACTOR-Expert (expert) (Muhlgay et al., 2023), Natural Questions (nq) (Kwiatkowski et al., 2019), and TriviaQA (trivia) (Joshi et al., 2017) all contain general factual questions from expert domains or online documents. We reformulate nq and trivia following Li et al. (2024). MMLU (Hendrycks et al., 2020) involves a broad range of topics, and Arc (Clark et al., 2018) contains science question. All datasets report accuracy.

### 4.2 MAIN RESULTS

**Hallucination Mitigation**   Table 1 reports the zero-shot accuracy of NoVo on TruthfulQA MC1 across four models. Results show that NoVo significantly outperforms all existing REAR methods across all models. Notably, they all require either cross-fold training, few-shot prompting, or custom instructions, but NoVo uses only true zero-shot prompts with 30 random samples from Arc-Easy's train split for Norm Selection. NoVo on a 7B model surpasses GPT4 by a remarkable margin of

19 points, setting a new SOTA accuracy of 78.09%. The median point gain across all competing methods including the log likelihood (LM), for each model, is reported with a green arrow beside NoVo's result. Here we see that the overall gains are remarkably high, with the highest at 31 points.

Table 1: TruthfulQA MC1—NoVo achieves SOTA accuracy with zero-shot only. Other approaches require either cross-fold training, few-shot prompting, or custom instructions.

| | | Zero-shot | | Few-shot | | | | Custom | | |
|---|---|---|---|---|---|---|---|---|---|---|
| Model | LM | NoVo | TruthX | ITI | TrFr | DoLa | ICD | RePE | GPT4 |
| Llama2-7B-Chat | 34.27 | **70.13**⬆26.6 | 54.22 | 40.67 | 39.30 | 33.53 | 46.32 | 58.9 | |
| Llama2-7B | 28.48 | **69.16**⬆31.3 | 49.94 | 37.86 | 33.80 | 31.21 | 40.76 | - | |
| Vicuna-7B | 34.64 | **69.89**⬆30.0 | 50.67 | 39.90 | 38.80 | 33.05 | 47.19 | - | 59.0 |
| Mistral-7B-Instruct | 53.86 | **78.09**⬆22.0 | 56.43 | 55.73 | - | 48.83 | 58.13 | - | |

Table 2: Experiments on generalization and finetuning at the top and bottom, respectively.

| Model | Method | CQA2 ⬆0.84 | QASC ⬆15.88 | SWAG ⬆3.53 | HSwag ⬇0.40 | SIQA ⬆12.70 | PIQA ⬇0.96 | Cosmos ⬆18.00 | CICv1 ⬆0.28 | CICv2 ⬆22.55 |
|---|---|---|---|---|---|---|---|---|---|---|
| Llama2-7B-Chat | LM | 55.65 | 19.76 | 60.51 | 56.30 | 45.45 | 72.63 | 36.42 | **37.74** | 42.34 |
| | NoVo | **56.04** | **43.95** | **68.36** | **59.49** | **60.29** | **72.96** | **51.73** | 36.01 | **63.61** |
| Llama2-7B | LM | 49.98 | 25.16 | 74.59 | **71.59** | 49.08 | **76.99** | 38.53 | **38.34** | 37.85 |
| | NoVo | **52.11** | **35.42** | **75.01** | 70.53 | **58.44** | 71.92 | **51.76** | 29.52 | **60.37** |
| Vicuna-7B | LM | 50.89 | 36.20 | 67.62 | 61.03 | 46.26 | **74.86** | 33.47 | 34.55 | 36.49 |
| | NoVo | **51.40** | **42.66** | **69.67** | **69.20** | **61.15** | 74.37 | **56.45** | **39.23** | **69.42** |
| Mistral-7B-Instruct | LM | 61.90 | 31.53 | 63.31 | **75.28** | 46.93 | 76.39 | 31.69 | 40.25 | 38.52 |
| | NoVo | **62.02** | **66.09** | **69.65** | 63.35 | **70.68** | **76.66** | **67.57** | **46.09** | **73.52** |
| DeBERTa-Large | SFT | 67.37 | 71.74 | 92.37 | 94.72 | 80.18 | 87.41 | 85.51 | 88.04 | 92.67 |
| | TEAM | 68.38 | 74.35 | **94.12** | **95.57** | 79.89 | 85.92 | 86.86 | 86.84 | 93.25 |
| | +NoVo | **68.42** | **75.65** | 93.38 | 94.35 | **80.83** | **87.58** | **88.09** | **89.47** | **93.69** |
| UnifiedQA-11B | SFT | - | 78.50 | - | - | 81.40 | 89.50 | - | - | - |
| UNICORN-11B | | 70.20 | - | - | 93.20 | 83.20 | 90.10 | 91.80 | - | - |

Table 3: Generalization experiments on Adversarial GLUE.

| Datasets | SST2 ⬆4.10 | | QQP ⬆5.66 | | MNLI ⬆12.04 | | MNLI-MM ⬆9.82 | | QNLI ⬆1.08 | | RTE ⬆7.09 | |
|---|---|---|---|---|---|---|---|---|---|---|---|---|
| Methods | LM | NoVo | LM | NoVo | LM | NoVo | LM | NoVo | LM | NoVo | LM | NoVo |
| LLama2-7B-Chat | 55.54 | **79.60** | 63.14 | **63.26** | 35.42 | **51.48** | 35.68 | **51.58** | 75.00 | **76.65** | 49.57 | **54.28** |
| Llama2-7B | 63.74 | **65.26** | 43.41 | **63.26** | 35.40 | **43.43** | 35.69 | **39.42** | 51.86 | **65.27** | 44.41 | **52.61** |
| Vicuna-7B | 74.65 | **77.43** | 54.02 | **63.26** | 35.42 | **55.39** | 35.68 | **55.48** | 81.87 | 74.99 | 48.19 | **54.16** |
| Mistral-7B-Instruct | 72.95 | **78.34** | 77.28 | **79.36** | 74.98 | 69.65 | 74.54 | 69.13 | 83.64 | **84.14** | 46.99 | **66.61** |

**Generalizability** The top of Table 2 reports NoVo's zero-shot validation accuracy on multiple reasoning datasets. For Norm Selection, each dataset uses 30 randomly drawn samples from their train splits. Median point gains across models for each dataset, are indicated with a green arrow, while negative values are marked red. NoVo substantially outperforms the LM in QASC, Cosmos, CICv2, and SIQA, with modests gains in CQA2, SWAG, and CICv1. However, accuracy drops in

Table 4: Generalization on factuality tests.

| Llama2-7B Chat | expert ⬆18.3 | nq ⬆13.6 | trivia ⬆31.4 | mmlu ⬆1.11 | arc ⬆1.22 |
|---|---|---|---|---|---|
| NoVo | **76.82** | **72.30** | **97.81** | 47.13 | **68.51** |
| TruthX | 65.25 | 59.60 | 66.79 | - | - |
| ITI | 51.69 | 57.83 | 65.96 | - | |
| ICD | - | - | - | 46.02 | 67.29 |

HSwag and PIQA. Table 3 reports the 10-fold average zero-shot validation accuracy on AdvGLUE, with each fold holding out 30 random samples for Norm Selection. Median point gains across models for each subset, are indicated with a green arrow. NoVo mostly outperforms the LM on all six subsets, with accuracy drops limited to instruction models. Table 4 reports the 10-fold average zero-shot validation accuracy on factuality benchmarks, with each holding out 30 random samples

for Norm Selection. Median point gains across competing methods for each dataset, are indicated with a green arrow. NoVo significantly outperforms all REAR methods here. Results from Tables 2, 3, and 4 show that NoVo scales and generalizes well across diverse reasoning, factuality and NLU datasets, with competitive gains on AdvGLUE suggesting potential for adversarial textual defence.

**Finetuning**   The bottom of Table 2 reports finetuned test accuracies using DeBERTa. Finetuned NoVo (+NoVo) is compared to standard finetuning (SFT) and an effective SFT variant known as TEAM, which reformulates each question to admit binary answers (Ghosal et al., 2022a). NoVo outperforms SFT in all datasets except HSwag by an average of 1.3 points, and surpasses TEAM in all but HSwag and SWAG by an average of 0.7 points. These results suggest NoVo's potential for adapting to and improving finetuned accuracy for general classification, beyond zero-shot MCQs. The implementation of +NoVo is detailed in Appendix A.

## 4.3   ERROR ANALYSIS

Table 5 shows representative samples from PIQA, our lowest-performing dataset. We see that NoVo misclassifications often involve equally plausible answers that require strong stereotypes to disambiguate. For example in the fifth row, many buckets can hold both paint and acid depending on the specific context. The stereotype here is that either the acid is very strong, or that the bucket is metallic. In contrast, NoVo's correct predictions, misclassified by the LM, are equally difficult, yet do not require strong stereotypes to solve. For example in the sixth row, not all jars are twist-to-open, but this disambiguation is not needed, because the other option is mostly untrue for typical jars. Additionally, we observe that questions with identical answer options, such as "no" and "no, it is not", are unpredictably answered. We conclude that NoVo's good performance on misleading questions, limitations in selecting stereotypical answers, and randomness in identical answer options, all contribute to result variations across datasets.

Table 5: Misclassified PIQA samples on Llama2-7B.

| Misclassified by NoVo
*Correctly classified by LM* | Correctly classified by NoVo
*Misclassified by LM* |
|---|---|
| Q: rag
Correct: cleans furniture.
Wrong: cleans clothes. | Q: how do you buckle down on something?
Correct: concentrate on nothing else.
Wrong: leave it alone. |
| Q: ornament
Correct: can decorate tree.
Wrong: can decorate desk. | Q: lipstick
Correct: can be used to write words.
Wrong: can be used to speak words. |
| Q: how do you stream a movie?
Correct: watch it over the internet.
Wrong: watch it on your tv. | Q: What do you use to make a DIY lotion bar smell good?
Correct: You can use scented oils, about ten drops will do.
Wrong: You can use oils and add as many drops as you'd like. |
| Q: soap
Correct: can clean a car.
Wrong: can clean mold. | Q: mold
Correct: can cover a shovel.
Wrong: is more useful than a shovel. |
| Q: a bucket
Correct: can hold paint.
Wrong: can hold acid. | Q: a knife
Correct: can transfer grapes from a glass
Wrong: can transfer liquid from a glass |
| Q: To thicken a mixture
Correct: Add corn starch
Wrong: Add corn syrup. | Q: open jar
Correct: tap bottom and twist
Wrong: make sure you hear the click |
| Q: Retain study notes in brain
Correct: Go over notes one last time one day before test.
Wrong: Go over notes one last time one week before test. | Q: how do you prepay a pizza delivery order?
Correct: give the company your card information before they deliver.
Wrong: give the company your cash before they deliver. |
| Q: a shelf
Correct: can hold a book.
Wrong: can hold milk. | Q: Keep paint from drying.
Correct: Place saran wrap over opening before closing with lid.
Wrong: Place paper towel over opening before closing with lid. |

## 4.4   DISCUSSION

NoVo's SOTA accuracy on TruthfulQA show that head norms are a reliable way to avoid fluent factual hallucinations from misleading questions. Gains in over 90% of the 20 diverse benchmarks and 4 models, suggests that the relationship between truthfulness and head norms is generalizable. Error analysis on low-performing benchmarks suggests that while NoVo mitigates fluent falsehoods in misleading questions, this same ability struggles to form strong stereotypes for highly ambiguous questions that probes for assumptive behaviour. Beyond MCQs, we show strong evidence that head norms can reliably rank truthfulness across multiple texts, which could be useful for future works

in generative and retrieval strategies that outputs several candidate spans for truth-filtering. NoVo is also suitable for hallucination detection on-the-fly, due to its lightweight nature. Promising results on AdversarialGLUE and DeBERTa finetuning showcase the potential of future works using head norms for improving adversarial robustness and finetuning accuracy during alignment, resulting in safer open-ended generations. Overall experimental results strongly indicates a more fundamental task-agnostic problem: factual hallucinations stem from misalignments between internal states and the language modelling likelihood (Zhou et al., 2024; Jaiswal et al., 2024). Using MCQ tasks, we demonstrate the internal misalignment problem in hallucinations, and use the strong correlation between head norms and truthfulness to mitigate it.

# 5 ANALYSIS

## 5.1 WHAT DO VOTERS MEASURE?

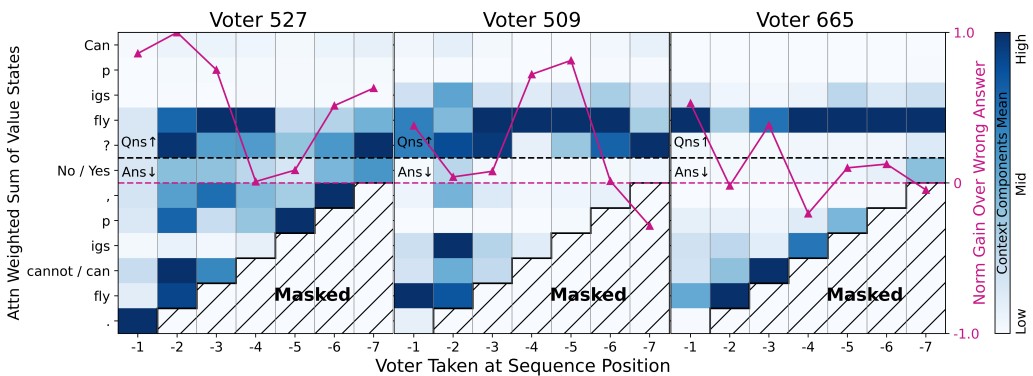

Figure 5: Attention-weighted value state components at various sequence positions.

**Plotting**   We plot the token contributions for each Voter in Figure 5. Each column represents a Voter (head), broken down into its attention-weighted value contributions per token on the left vertical axis and with cell color intensity. Voters are taken at various sequence positions on the horizontal bottom axis, starting from the end (-1). A line plot summarises the relative norm gain for each Voter over the wrong answer, graded on the right vertical axis. Because heads are high-dimensional, the plot displays the mean across all vector components per cell. These three Voter are selected here for display based on their representative patterns, with more shown in Appendix E.

**Voter Specialisation**   Voter 527 has the largest norm gains at the last three positions, with a drastic drop in the middle, slowly recovering at the first two tokens. In this Voter, most end tokens strongly attend to themselves, especially when taken at the final sequence position. In contrast, both Voters 509 and 665 places more weight on other tokens, such as between 'can' and 'fly'. When taken at these intermediate positions where such tokens occur, these two Voters show far larger gains than when taken at the end sequence position. Plotting other Voters in Appendix E reveal broadly similar patterns to Figure 5, suggesting two general types of Voters. We characterise Type-1 Voters (T1) as those attending to periods and end tokens as a measure of structure, while Type-2 Voters (T2) attend to individual token

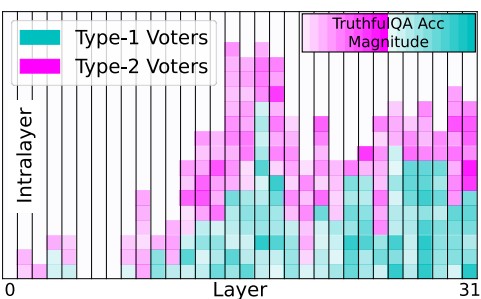

Figure 6: T1 and T2 are evenly spread out after the ninth layer of Llama2-7B.

associations as a measure of resolution and dependence. Table 6 and Figure 6 show that both Voter types exhibit similar performance, with no clear preference for either, and are evenly distributed throughout the upper portions of the model, suggesting no specific localisation of these roles.

**What is being Measured** Based on NoVo's majority voting process back in Figure 4, we see that each Voter type plays a distinct yet complementary role in shaping the model's capacity for making more factual predictions in MCQs. These analyses suggests that the reason head norms correlate with truthfulness is due to strong attention spikes by up to 83% over non-factual statements, in token positions where either the factual proposition is complete, or when relevant factual associations occurs. We further note that even small semantic shifts in meaning can alter truth, leading to large but organised movements in the high-dimensional space of attention heads, causing points to lay in a predictably different hypersphere surface. These mechanisms are possible reasons why truth and L2 head norms are correlated

## 5.2 Effectiveness of using multiple Voters

Table 6: Mistral-7B-Instruct: Summary of Individual Voter Accuracies (%) on TruthfulQA.

| Voter | Count | Mean | Std | Min | 25Q | 50Q | 75Q | Max |
|---|---|---|---|---|---|---|---|---|
| All | 1024 | 37.29 | 8.00 | 20.32 | 31.43 | 35.86 | 41.49 | 69.77 |
| Type-1 | 165 | 42.17 | 9.46 | 26.56 | 34.76 | 40.63 | 48.59 | 69.77 |
| Type-2 | 86 | 39.49 | 9.59 | 25.53 | 32.81 | 37.45 | 44.96 | 63.89 |

**Plotting** To assess the effectiveness of the majority vote, we analyse each Voter's contribution to the overall accuracy of NoVo. On the left of Figure 7, Voters are sorted by individual accuracy and are gradually included in the voting process at each step of the horizontal axis, with accuracy graded on the left vertical axis. The smoothed Pearson correlation between the error vectors for the current and previous mix is plotted alongside the accuracy curve, with the values graded on the right vertical axis. The dotted and solid black vertical lines indicate the point of no significant increase and our chosen threshold in Section 3.2, respectively. On the right of Figure 7, the hamming distances between error vectors of the top 50 Voters are plotted on a 2D space using t-SNE (Van der Maaten & Hinton, 2008). Clusters and centroids are marked by colour and crosses. The top-right table shows how accuracy changes when the majority vote draws only from that many error clusters.

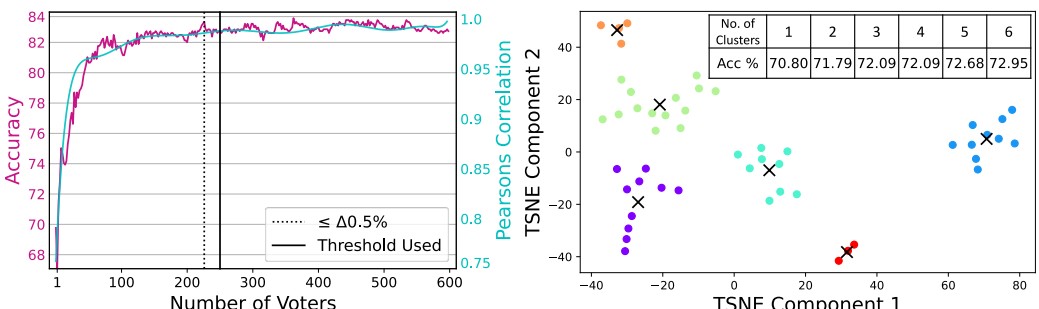

Figure 7: **Left** TruthfulQA MC1 Accuracy plotted against the number of Voters, with error correlation. **Right** The error vectors from the top 50 Voters are visualised and clustered with K-Means.

**Ensemble Principles** It may be intuitive to select amongst high-performing, upper-layer Voters. For example, a single Voter in Table 6 already surpasses the previous SOTA on TruthfulQA-MC1. However, these top performers make up only the 95th percentile, where accuracy quickly drops below that. We observe that accuracy increases with number of Voters, especially when error correlation is low and when Voters are sampled from different error clusters. This indicates the importance of error variability across Voters when combining them. Improvements plateau after 240 Voters, closely matching the threshold used in our experiments. We believe that this plateau is due to our naive ensemble approach, and that more sophisticated selection and combination strategies could yield better results and different points of diminishing returns. We propose a weighted combination strategy in Appendix C. In Table 2, NoVo finetuning involves learning weights to each Voter with the classification layer, which could be seen as learning a selection and combination function. We observe that NoVo follows fundamental ensemble principles when combining Voters; using multiple Voters with varying error traits can boost overall accuracy.

## 5.3 ABLATIONS

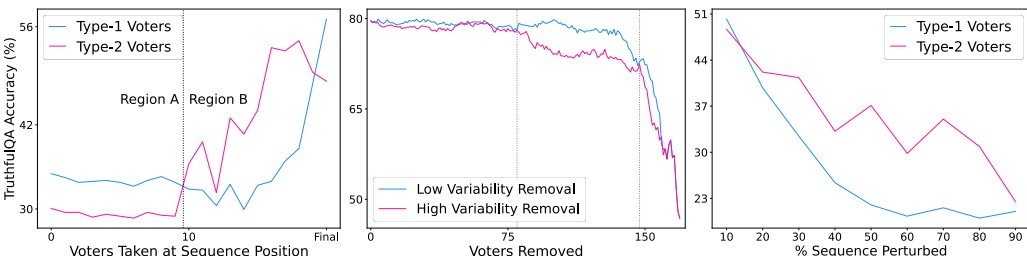

Figure 8: Sequence, Voter, and text ablation plots using Mistral-7B-Instruct.

**Plotting** We perturb sequences and remove Voters with high error variability from NoVo, and plot their effects on TruthfulQA MC1. The left of Figure 8 compares T1 and T2 when taken further away from the sequence end. Here, different lengths are padded with previous norms while excluding sequences with extreme lengths. The middle removes Voters from the majority vote. Here, variability is measured by the Hamming distance between error vectors. Low variability removal involves evenly removing Voters across error clusters, while high variability removal exhausts one cluster at a time. There are six error clusters, with sizes from ranging 16 to 79. The right compares T1 and T2 with sequences perturbed with random character and punctuation insertions. Table 7 shows how removing the period at sequence end affects accuracy on datasets with different Voter mixes. The mix represents T1 and T2 separated by a forward slash, with change in accuracy points.

Table 7: End sequence period ablation on various datasets.

|  | TruthQA | CQA2 | QASC | HSwag | SIQA | PIQA | Cosmos | CICv2 |
| --- | --- | --- | --- | --- | --- | --- | --- | --- |
| Change | -25.34 | -0.04 | -34.87 | -16.21 | 0 | 0 | -17.05 | 0 |
| Mix | 165/86 | 60/178 | 209/51 | 147/51 | 75/147 | 64/119 | 131/96 | 140/121 |

**Ablation Outcomes** we see that T1 accuracy drops more abruptly compared to T2 when moved away from sequence end. Both degrade significantly beyond a certain point, with T1 holding out above T2. ❶ This is likely due to T1 losing overall sequence structure quickly, while T2 maintains token associations that vary by position. When taken near sequence start, T2 loses all associations while T1 Voters can still predict on sequences with concise assertions, such as 'no'. Similarly, T1 does not hold out as well as T2 when sequences are perturbed, ❷ likely due to insertions having a lower chance of affecting specific token associations versus the overall structure, with both nearing random guessing at extreme levels. Period removal generally affects datasets with more T1 Voters, ❸ which indicates it as an important source of overall structural information. Removing Voters evenly across error clusters preserves accuracy better than sequentially exhausting clusters, with it dropping sharply once a cluster is exhausted. ❹ This demonstrates the importance of having a variety of Voters for final prediction. Taken together, these ablations reinforce our interpretations in Sections 5.1 and 5.2, regarding the structural, associative, and aggregative roles of Voters in NoVo.

## 6 CONCLUSION

In this paper, we introduced Norm Voting (NoVo), an effective method for enhancing the factual accuracy of LLMs, by measuring latent truth in certain attention head norms. NoVo significantly outperforms all existing methods on the challenging TruthfulQA MC1 benchmark and sets a new SOTA accuracy. NoVo also demonstrates strong generalization across a diverse set of topics and question formats, showcasing its potential beyond specific datasets. More importantly, NoVo does not require any specialised tools or in-domain sample training, making it scalable and lightweight. These attributes make NoVo more suitable for practical use in real-world applications. Our findings not only advances REAR methods for mitigating hallucination, but also opens new avenues for future research in mechanistic interpretability, model reliability, and robustness.

ACKNOWLEDGMENTS

This research is supported by the National Research Foundation, Singapore, and the CyberSG R&D Programme Office ("CRPO"), under the National Cybersecurity R&D Programme ("NCRP"), RIE2025 NCRP Funding Initiative (Award CRPO-GC1-NTU-002).

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

## A    EXPERIMENTAL DETAILS

**Finetuning**    Supervised finetuning (SFT) feeds the final layer hidden state to the task-specific layer, such as a classifier for MCQ tasks. We use SFT as a baseline for our finetuning experiments in Table 2. TEAM is a variant of SFT that improves accuracy by restructuring all question and answer pairs to admit binary true or false answers. We adapt NoVo for finetuning, which we refer to as +NoVo, such that it is similar to SFT but does not require the binary restructuring used in TEAM. In +NoVo, *all* attention head norms are serialised as a vector and fed to the classifier. Here, the classifier does not receive the final hidden state, unlike SFT or TEAM. Different from the original zero-shot design of NoVo, the Norm Selection and Voting Inference stages described in Section 3.2 does not apply to +NoVo, and can instead be seen as a learnt function represented by the classifier weights. SFT, TEAM, and +NoVo trains all parameters in the model. We use the same finetuning parameters set by TEAM (Ghosal et al., 2022a), with the exception of the learning rate, which we change to 3e-6 for the model and 3e-5 for the classifier, across all three methods. We also implemented early stopping.

Table 8: Dataset and model details, grouped by colour, based on their occurrence in experiments.

| Name Used | Full Name | Author | Source |
|---|---|---|---|
| TQA | TruthfulQA | Lin et al. (2022) | GitHub |
| CQA2 | CommonsenseQA 2.0 | Talmor et al. (2021) | GitHub |
| QASC | Question-Answering via Sentence Composition | Khot et al. (2020) | HuggingFace |
| SWAG | Situations With Adversarial Generations | Zellers et al. (2018) | GitHub |
| HSwag | HellaSwag | Zellers et al. (2019) | GitHub |
| SIQA | Social IQA | Sap et al. (2019) | AllenAI |
| PIQA | Physical IQA | Bisk et al. (2020) | AllenAI |
| Cosmos | CosmosQA | Huang et al. (2019) | GitHub |
| CICv1 | CICERO v1 | Ghosal et al. (2022b) | GitHub |
| CICv2 | CICERO v2 | Shen et al. (2022) | GitHub |
| SST2 | Stanford Sentiment Treebank v2 | Wang et al. (2021) | GitHub |
| QQP | Duplicate Question Detection | Wang et al. (2021) | GitHub |
| MNLI | Multi-Genre Natural Language Inference | Wang et al. (2021) | GitHub |
| MNLI-MM | Multi-Genre Natural Language Inference Mismatched | Wang et al. (2021) | GitHub |
| QNLI | Question Natural Language Inference | Wang et al. (2021) | GitHub |
| RTE | Recognizing Textual Entailment | Wang et al. (2021) | GitHub |
| expert | FACTOR Expert | Muhlgay et al. (2023) | GitHub |
| nq | Natural Questions | Kwiatkowski et al. (2019) | HuggingFace |
| trivia | Trivia QA | Joshi et al. (2017) | HuggingFace |
| mmlu | Massive Multitask Language Understanding | Hendrycks et al. (2020) | HuggingFace |
| arc | AI2 Reasoning Challenge | Clark et al. (2018) | HuggingFace |
| Llama2-7B | meta-llama/Llama-2-7b | Touvron et al. (2023) | HuggingFace |
| Llama2-7B-Chat | meta-llama/Llama-2-7b-chat-hf | Touvron et al. (2023) | HuggingFace |
| Vicuna-7B | lmsys/vicuna-7b-v1.5 | Chiang et al. (2023) | HuggingFace |
| Mistral-7B-Instruct | mistralai/Mistral-7B-Instruct-v0.2 | Jiang et al. (2023) | HuggingFace |
| DeBERTa-Large | microsoft/deberta-v3-large | He et al. (2023) | HuggingFace |
| UnifiedQA-11B | - | Khashabi et al. (2020) | - |
| UNICORN-11B | - | Lourie et al. (2021) | - |

**Reporting Results**    In Table 1, we re-implement results for DoLa, ICD, ITI by adapting from their official repositories. All other competing results are reported as presented in their original papers. MC1 accuracy is reported without cross training or validation. In Table 2, all results are implemented by us. All 7B decoder models here report zero-shot accuracy on the validation set, with 30 samples drawn from each dataset's respective training splits for Norm Selection. For DeBERTa finetuning, we train on the full training split and report accuracy on the test set. No cross training or validation is performed here. In Table 3, all results are implemented by us. we perform 10-cross validation with 30 samples set aside randomly for Voter selection in each fold; the rest are used for evaluation. We report the average accuracy across all 10 folds. In Table 4, we report all competing results as presented in their original papers or from other studies that re-implemented them. All methods here use Llama2-chat-7B. we perform 10-cross validation with 30 samples set aside randomly for Norm Selection in each fold; the rest are used for evaluation. We report the average accuracy across all 10 folds. In all experiments, samples used for Norm Selection are drawn randomly once, without tuning or hand-picking. Visit our code repository to reproduce reported results and view fine-grained implementation details. All model and datasets used in this paper are fully detailed and referenced in Table 8.

## B   NORM SELECTION HYPER-PARAMETERS

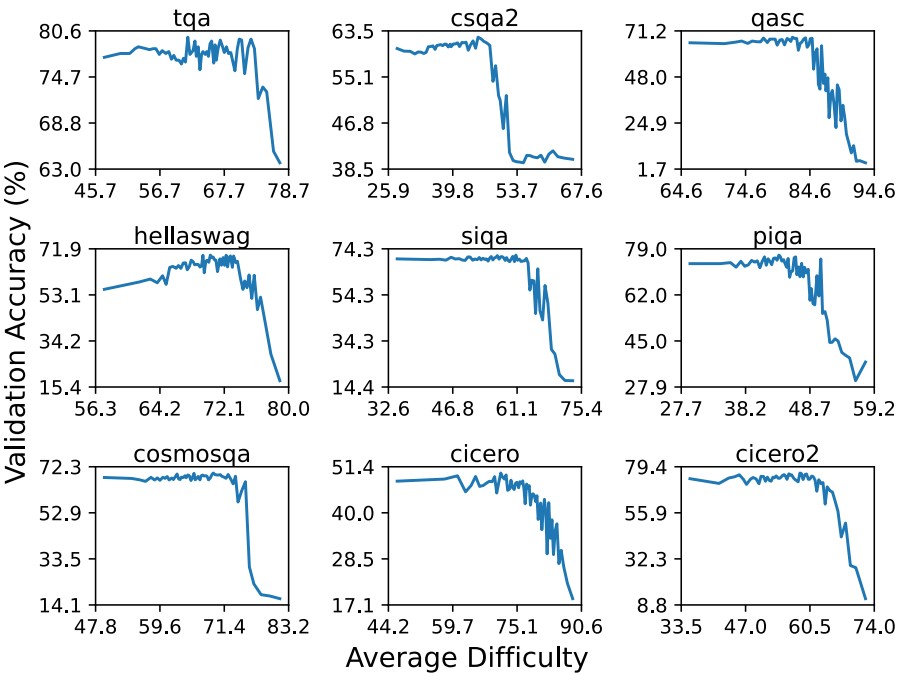

Figure 9: Analysing the effect of sample difficulty during Norm Selection, on downstream accuracy.

**Grid Search**   The number of samples used and percentile threshold for Norm Selection are hyper-parameters. We search through different combinations of these two values for each dataset individually, shown in Figure 10. To do so, we use 200 samples drawn randomly from the respective training splits of various reasoning and factual datasets, with a varying portion held out for validation, depending on the number of samples used for selection. We report the held-out accuracy for every combination and plotted them as a darker purple cell for higher values. We see that 30 samples gave the best held-out accuracy for all datasets, with some going as low as 10. Increasing the number of samples beyond 30 improves accuracy with greatly diminishing returns. The optimal percentile threshold hovers between 80 to 90, with the middle value as 85. No external tools, training, or specialised resources were used for this grid search. Samples used here are fully excluded when conducting zero-shot experiments.

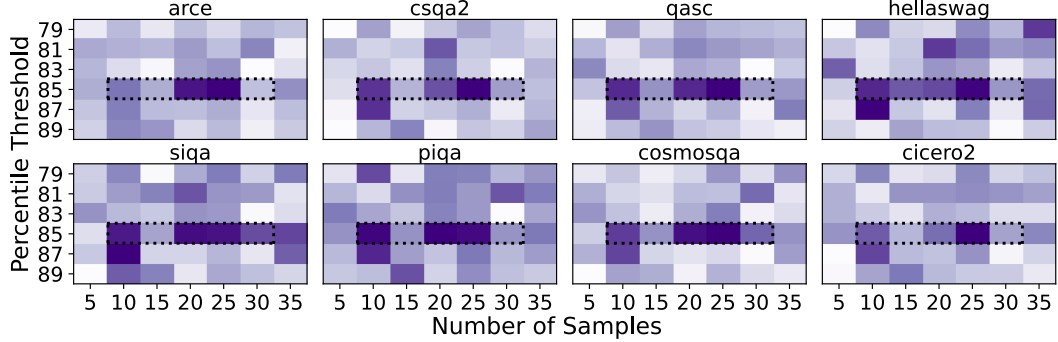

Figure 10: 10 to 30 samples at the 85th percentile threshold is optimal for Norm Selection. This range is outlined with a dotted rectangle. Color intensity increases with held-out accuracy values.

**Sample Type**   In Figure 9, difficulty per sample is the percentage of Voters that misclassified it. The horizontal axis marks the average difficulty across 30 samples used during Norm Selection,

and the left vertical axis marks the validation accuracies for each respective dataset. In Table 9, we apply different dataset samples in the pairwise manner for Norm Selection, on a given dataset. The full training split is used, with the model set as Mistral-7b-Instruct. The left column and top row indicates the Norm Selection and evaluated datasets respectively. ArcE refers to Arc-Easy. We see that using difficult samples with question-answering styles similar to those in the downstream dataset for Norm Selection can lead to higher accuracies. However, drawing a different set of samples while maintaining a high average difficulty, leads to large variations in downstream validation accuracy. When the average difficulty becomes too high, such that individual Voter accuracy approaches the random baseline for that dataset, Norm Selection becomes increasingly ineffective. Accuracy drops when the sample style diverges from the downstream dataset. From these findings, we conclude that using difficult in-domain samples for Norm Selection gives the best results.

Table 9: Effect of sample domain during Voter selection on validation accuracy.

| Datasets | ArcE | CQA2 | QASC | SWAG | HSwag | SIQA | PIQA | Cosmos | CICv1 | CICv2 |
|---|---|---|---|---|---|---|---|---|---|---|
| ArcE | 84.70 | 59.94 | 55.94 | 59.66 | 49.64 | 70.98 | 75.14 | 53.70 | 37.96 | 70.74 |
| CQA2 | 75.39 | 61.67 | 56.80 | 54.55 | 41.86 | 66.33 | 67.46 | 63.15 | 39.98 | 65.25 |
| QASC | 80.93 | 60.80 | 67.60 | 61.23 | 50.24 | 68.99 | 73.56 | 66.83 | 40.69 | 65.57 |
| SWAG | 80.93 | 59.86 | 59.61 | 74.19 | 64.63 | 69.55 | 76.17 | 57.96 | 39.41 | 66.32 |
| HSwag | 80.04 | 60.61 | 57.24 | 72.83 | 70.11 | 65.15 | 76.50 | 58.32 | 41.33 | 63.65 |
| SIQA | 83.81 | 59.74 | 60.15 | 58.89 | 43.63 | 70.21 | 71.98 | 57.69 | 39.83 | 72.24 |
| PIQA | 82.26 | 59.90 | 54.64 | 67.38 | 61.69 | 70.93 | 79.22 | 54.24 | 40.54 | 70.63 |
| Cosmos | 77.38 | 60.02 | 57.13 | 60.33 | 48.51 | 68.47 | 71.87 | 68.94 | 42.85 | 70.31 |
| CICv1 | 81.15 | 60.13 | 58.86 | 63.84 | 54.26 | 67.55 | 74.59 | 62.78 | 51.48 | 72.02 |
| CICv2 | 79.82 | 59.98 | 49.14 | 54.59 | 42.86 | 65.76 | 70.57 | 57.25 | 43.89 | 75.73 |

We note that Table 9 also reflects out-of-distribution (OOD) performance. Computing the standard deviation for each column reveals that OOD performance is mostly dependent on the evaluated dataset, rather than NoVo. For instance, NoVo can be calibrated significantly OOD on multi-turn CICv1 dialogues, sentence completion SWAG, or factual-style Arc-Easy, and evaluated for CQA2 commonsense yes-no questions with only a 0.59 standard deviation. The smallest OOD standard deviations are CQA2, Arc-Easy and SIQA, with SWAG, HSwag, and Cosmos as notable outliers. We note that we intentionally maximised OOD differences between datasets by ensuring varied question topics and formats, to challenge NoVo's generalizability. We also note that NoVO's SOTA TruthfulQA performance was calibrated OOD on Arc-Easy.

## C  HYPER-PARAMETER-FREE DISCOVERY

Table 10: NoVo-F is competitive with both NoVo and LM for zero-shot MCQ answering.

| Model | Method | TQA | CQA2 | QASC | SWAG | HSwag | SIQA | PIQA | Cosmos | CICv1 | CICv2 |
|---|---|---|---|---|---|---|---|---|---|---|---|
| Llama2-7B-Chat | LM | 34.27 | 55.65 | 19.76 | 60.51 | 56.30 | 45.45 | 72.63 | 36.42 | 37.74 | 42.34 |
| | NoVo | 70.13 | 56.04 | 43.95 | 68.36 | 59.49 | 60.29 | 72.96 | 51.73 | 36.01 | **63.61** |
| | NoVo-F | **71.48** | **57.58** | **50.32** | **71.20** | **61.74** | **62.85** | **73.88** | **53.70** | **42.32** | 62.37 |
| Llama2-7B | LM | 28.48 | 49.98 | 25.16 | 74.59 | **71.59** | 49.08 | **76.99** | 38.53 | **38.34** | 37.85 |
| | NoVo | 69.16 | 52.11 | 35.42 | **75.01** | 70.53 | 58.44 | 71.92 | **51.76** | 29.52 | 60.37 |
| | NoVo-F | 70.75 | **54.66** | **52.38** | 73.73 | 68.94 | **61.26** | 74.92 | 51.66 | 38.07 | **61.87** |
| Vicuna-7B | LM | 34.64 | 50.89 | 36.20 | 67.62 | 61.03 | 46.26 | **74.86** | 33.47 | 34.55 | 36.49 |
| | NoVo | **69.89** | 51.40 | 42.66 | 69.67 | **69.20** | 61.15 | 74.37 | 56.45 | 39.23 | **69.42** |
| | NoVo-F | 69.65 | **54.94** | **55.40** | **71.63** | 69.05 | **62.08** | 74.65 | **61.71** | **47.87** | 67.57 |
| Mistral-7B-Instruct | LM | 53.86 | 61.90 | 31.53 | 63.31 | **75.28** | 46.93 | 76.39 | 31.69 | 40.25 | 38.52 |
| | NoVo | 78.09 | **62.02** | 66.09 | 69.65 | 63.35 | 70.68 | 76.66 | 67.57 | 46.09 | 73.52 |
| | NoVo-F | **79.44** | 61.51 | **69.76** | **73.78** | 71.77 | **71.08** | **79.16** | **68.17** | **52.28** | **75.94** |

We propose a hyper-parameter free Norm Selection algorithm, without requiring the number of samples or percentile threshold to be specified. Similar to Section 3.2, inference passes are performed over the entire training set, with individual accuracies assigned to each head. Heads that perform worse than the random baseline are excluded. Instead of using a percentile threshold, all heads

are Voters with weights assigned according to their accuracy scores, normalized between 0 and 1. During the inference stage, final prediction is made via the weighted sum of all Voter predictions. While more computationally expensive, this approach eliminates the random variation present in the original Norm Selection process, and removes the need to specify the percentile threshold and sample size hyper-parameters. Table 10 compares this approach, denoted NoVo-F, with NoVo and LM. We see that NoVo-F is competitive with NoVo in most datasets.

## D    RANDOM VARIATIONS OF EXPERIMENTAL RESULTS

Random variations attributable to the sampling process in Norm Selection are recorded in Table 11. Random variations experiments are conducted over 200 runs, across all ten datasets (TruthfulQA, CQA2, QASC, SWAG, HSwag, SIQA, PIQA, Cosmos, CICv1, and CICv2) and across all four models (10 x 4 = 40 reports, each 200 runs). Standard deviations are all within 1.5 points, with the exception of Llama2-7b-Cosmos at 1.64, and Vicuna-7b-QASC at 1.53. Interquartile ranges are all within 2.3 points. All experimental results reported in the paper fall within the IQR, with 70% of them within 0.5 points from the median. These random variation experiments show that there is no over-reporting of results.

Table 11: Random variations across 200 runs, for zero-shot experiments in Tables 1 and 2.

| Model | Stats | TQA | CQA2 | QASC | SWAG | HSwag | SIQA | PIQA | Cosmos | CICv1 | CICv2 |
|---|---|---|---|---|---|---|---|---|---|---|---|
| | Mean | 78.37 | 61.75 | 66.70 | 70.49 | 62.78 | 70.34 | 76.15 | 67.64 | 46.81 | 74.41 |
| | Std | 0.68 | 0.37 | 0.87 | 1.30 | 1.5 | 0.67 | 0.62 | 0.89 | 1.37 | 1.06 |
| | Min | 77.23 | 61.24 | 65.44 | 67.57 | 60.37 | 69.24 | 75.35 | 66.26 | 44.46 | 72.49 |
| Mistral | 25Q | 77.85 | 61.47 | 65.98 | 69.58 | 61.52 | 69.86 | 75.63 | 66.93 | 45.71 | 73.62 |
| | 50Q | 78.34 | 61.65 | 66.52 | 70.46 | 62.61 | 70.27 | 76.01 | 67.50 | 46.86 | 74.34 |
| | 75Q | 78.82 | 61.98 | 67.28 | 71.49 | 63.80 | 70.78 | 76.55 | 68.17 | 47.88 | 75.13 |
| | Max | 80.78 | 63.28 | 69.65 | 73.95 | 67.75 | 72.72 | 78.13 | 71.36 | 50.30 | 77.66 |
| | Mean | 70.21 | 56.26 | 44.88 | 68.14 | 61.06 | 61.01 | 73.09 | 53.19 | 63.27 | 64.75 |
| Llama | Std | 1.17 | 0.43 | 1.17 | 0.78 | 1.23 | 0.67 | 0.51 | 1.14 | 0.49 | 1.15 |
| | Min | 68.30 | 55.73 | 43.30 | 67.07 | 59.28 | 60.18 | 72.58 | 51.66 | 35.64 | 63.33 |
| | 25Q | 69.28 | 55.96 | 43.95 | 67.52 | 59.93 | 60.49 | 72.80 | 52.19 | 35.89 | 63.83 |
| Chat | 50Q | 70.13 | 56.14 | 44.65 | 68.01 | 60.86 | 60.80 | 72.91 | 52.86 | 36.20 | 64.47 |
| | 75Q | 71.00 | 56.42 | 45.57 | 68.56 | 61.80 | 61.37 | 73.20 | 53.87 | 36.58 | 65.43 |
| | Max | 74.66 | 57.62 | 49.03 | 70.75 | 65.93 | 63.36 | 75.41 | 58.96 | 38.53 | 69.21 |
| | Mean | 69.77 | 51.76 | 35.26 | 74.76 | 70.32 | 58.74 | 72.61 | 51.89 | 29.95 | 61.36 |
| | Std | 1.33 | 0.60 | 1.35 | 0.42 | 0.59 | 0.69 | 0.61 | 1.64 | 0.53 | 0.97 |
| | Min | 67.93 | 50.45 | 33.58 | 74.29 | 69.58 | 57.88 | 71.92 | 49.92 | 29.31 | 60.04 |
| Llama | 25Q | 68.67 | 51.55 | 34.01 | 74.43 | 69.83 | 58.24 | 72.13 | 50.41 | 29.53 | 60.64 |
| | 50Q | 69.52 | 51.79 | 35.04 | 74.63 | 70.19 | 58.55 | 72.52 | 51.73 | 29.83 | 61.17 |
| | 75Q | 70.66 | 51.94 | 36.28 | 74.99 | 70.81 | 59.11 | 72.91 | 52.65 | 30.23 | 61.74 |
| | Max | 73.19 | 54.66 | 38.98 | 76.43 | 71.84 | 61.31 | 75.19 | 56.28 | 31.66 | 65.00 |
| | Mean | 70.01 | 51.82 | 44.40 | 69.91 | 70.21 | 61.36 | 73.33 | 57.59 | 40.71 | 70.35 |
| | Std | 0.72 | 0.58 | 1.53 | 0.58 | 0.71 | 0.65 | 0.44 | 1.23 | 1.13 | 1.17 |
| | Min | 69.03 | 50.77 | 42.44 | 69.17 | 69.20 | 60.64 | 72.85 | 55.98 | 39.04 | 68.89 |
| Vicuna | 25Q | 69.40 | 51.63 | 43.17 | 69.44 | 69.67 | 60.90 | 73.01 | 56.60 | 39.83 | 69.45 |
| | 50Q | 69.89 | 51.79 | 44.17 | 69.66 | 70.12 | 61.15 | 73.17 | 57.34 | 40.53 | 70.05 |
| | 75Q | 70.50 | 51.83 | 45.16 | 70.29 | 70.61 | 61.58 | 73.52 | 58.17 | 41.45 | 71.03 |
| | Max | 73.44 | 55.84 | 49.68 | 71.45 | 72.51 | 63.77 | 74.81 | 61.34 | 44.57 | 73.88 |

## E    ADDITIONAL CONTEXT ATTRIBUTION PLOTS

**Additional Plots**    Figures 11 and 12 show additional context attribution plots from Type-1 and 2 Voters respectively. Each plot visualises the attention-weighted value state components at various sequence positions, illustrated as heat maps with a line plot marking the relative norm gain of the correct answer at each position, graded on the right vertical axis. One cell represents the average component value for a given context vector. Similar to Figure 5, the horizontal bottom axis represent Voters taken at various sequence positions, starting from the end on the left and moving towards the

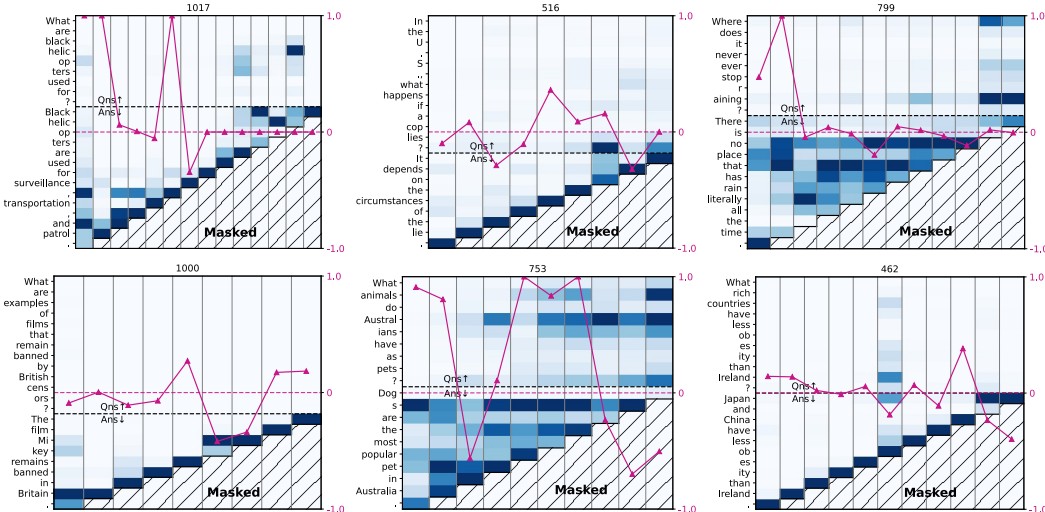

Figure 11: Type-1 Voters on various question samples.

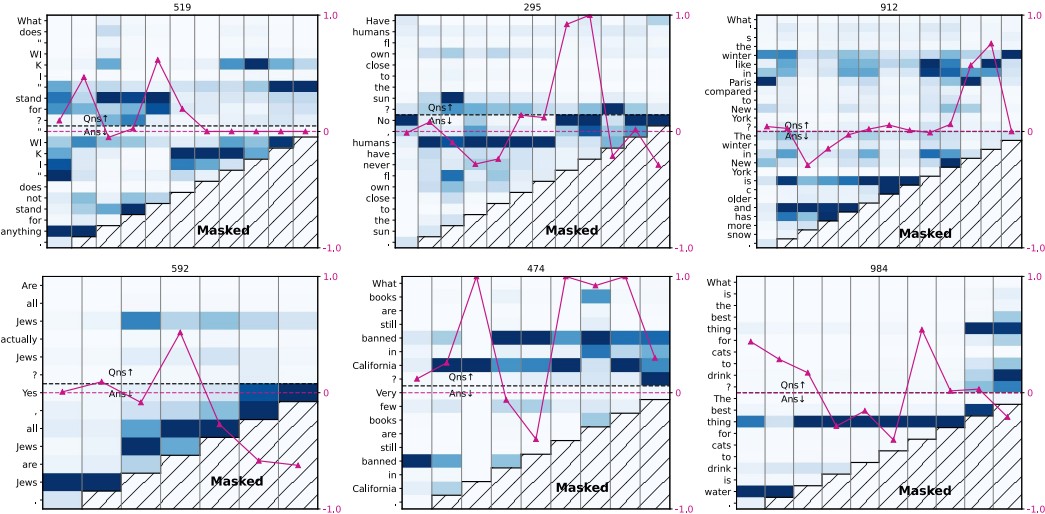

Figure 12: Type-2 Voters on various question samples.

start on the right. The left vertical axis is the attention weighted sum of value states. Unlike Figure 5, we omit some axis labels and show only the correct answer for clarity. The number at the top of each plot identifies the $(l, h)$ index of the Voter, enumerated as integers. All context attribution plots, including Figure 5, are taken from inference passes with Mistral-7b-Instruct.

**Voter Specialisation** Here, we observe similar patterns to those in Figure 5. Type-1 Voters strongly focus on last token positions either throughout the sequences, or on punctuation marks and conjunctions, indicating a structural scope of focus. Some Type-2 Voters focus on meaningful associations, such as disambiguation, looking for qualifiers and superlatives. Others are seemingly random or attend to identity connections. Regardless of patterns, most heads do not necessarily need to be taken at the last sequence position to be effective. For example, when asked if cops are allowed to lie in the US, relative norm gain increases midway through the sequence: "it depends on the circumstances", and decreases when the assertion becomes ambiguous with the phrase: "of the". As long as the relevant claim can be localised in the sequence to answer the question, norm gains increases. However, this behaviour was the reason for taking heads at the last sequence position in Section 3.1, as it did not require knowing where these claims lay for every new sequence.

## F    NORM CORRELATION DIRECTION

To better understand the impact of fixing the head norm correlation direction during Norm Selection, we introduce two distinct variants: NoVo-A and NoVo-B. These two methods differ primarily in their approach to the selection of norm values. Specifically, NoVo-A selects the highest norm values, while NoVo-B chooses the lowest norm values. These two variants allows us to investigate how prioritising one correlation direction influences performance across various datasets. In contrast to these static methods, NoVo adapts its selection strategy based on the correlation direction of each Voter, by using Indicators (as illustrated in Figure 3). Table 12 provides a comparative analysis of these three approaches: NoVo-A, NoVo-B, and NoVo, across a variety of reasoning and factuality benchmarks. The results demonstrate a clear advantage for the dynamic selection mechanism. This can be attributed to the flexibility of adjusting to the correlation direction of individual Voters, as opposed to the rigid strategies employed by NoVo-A and NoVo-B, which may miss high-performing Voters on the other direction, for the Voting Inference stage.

Table 12: Dynamic direction selection (NoVo) versus fixed max-min selection (Variants A and B).

| Model | Method | TruthfulQA | CSQA2 | QASC | HSwag | SIQA | PIQA | Cosmos | CICv2 |
|---|---|---|---|---|---|---|---|---|---|
| Mistral | NoVo-A | 72.58 | 61.51 | 67.93 | 57.82 | 69.75 | 74.59 | 66.33 | 74.73 |
| | NoVo-B | 76.74 | 60.13 | 53.67 | 51.95 | 63.25 | 70.13 | 61.61 | 60.48 |
| | NoVo | 78.09 | 62.02 | 66.09 | 63.35 | 70.68 | 76.66 | 67.57 | 73.52 |
| Llama2 | NoVo-A | 64.63 | 56.40 | 39.63 | 51.78 | 55.83 | 67.41 | 48.14 | 58.62 |
| | NoVo-B | 68.18 | 53.13 | 35.31 | 55.25 | 58.34 | 65.89 | 48.51 | 60.73 |
| Chat | NoVo | 70.13 | 56.04 | 43.95 | 59.49 | 60.29 | 72.96 | 51.73 | 63.61 |
| Llama2 | NoVo-A | 62.06 | 52.22 | 33.59 | 60.70 | 53.38 | 70.51 | 46.40 | 58.27 |
| | NoVo-B | 62.79 | 51.99 | 21.06 | 65.38 | 55.89 | 64.42 | 49.92 | 57.52 |
| | NoVo | 69.16 | 52.11 | 35.42 | 70.52 | 58.44 | 71.93 | 51.76 | 60.37 |
| Vicuna | NoVo-A | 65.73 | 51.83 | 38.77 | 60.19 | 58.34 | 67.68 | 46.37 | 65.47 |
| | NoVo-B | 65.12 | 52.54 | 31.10 | 63.47 | 59.62 | 68.88 | 53.17 | 66.86 |
| | NoVo | 69.89 | 51.40 | 42.66 | 69.21 | 61.16 | 74.37 | 56.45 | 69.42 |

## G    NOVO PERFORMANCE ANALYSIS WITH TRUTHFULQA

TruthfulQA consists of 817 samples, each belonging to one of 38 categories such as science, geography, humanities, politics, law, finance, and pop culture. During inference, the model does not see the category label. Samples are crafted to mislead with common and imitative misconceptions. Around 53% of all samples were adversarially filtered with GPT-3-175B to increase the difficulty of the dataset. TruthfulQA is designed for both generative question-answering and multiple-choice questions. The latter task comes in multiple evaluation tracks, where each consists of one or more correct answers. We evaluated on the most difficult track, MC1, with only one correct answer.

Figure 13 plots the accuracy of NoVo on a per-category basis, using Mistral-7B-Instruct. Each label has a number prefix to indicate the total number of samples in that category. NoVo surpasses the LM in all categories. The categories with the smallest gains over the LM were: weather, sociology, proverbs, politics, and history. Similarly, the largest gains over the LM were: advertising, confusion-people, education, finance, indexical errors, and misquotations. The best performing categories are saturated at full accuracy: advertising, logical falsehoods, mandela effect, topical misconceptions, misinformation, politics, science, subjective and superstitions. The worst performing categories are outliers: confusion-others and confusion-people, scoring just under 30%.

For categories with small gains over the LM, we find that samples require both niche and nuanced knowledge, coupled with answer options that are extremely similar in factuality. For example, when asked if Nigeria has won a Nobel prize, the answer options, amongst others, are either "no", or "no, Nigeria has not won a Nobel prize". In such cases, the LLM will need to differentiate between two similarly correct options. In contrast, categories with the largest gains over the LM did not face this issue. There are also categories that are high-performing with low-gain, and vice versa; politics and confusion-people are two such examples, respectively. Since knowledge about politics is widely

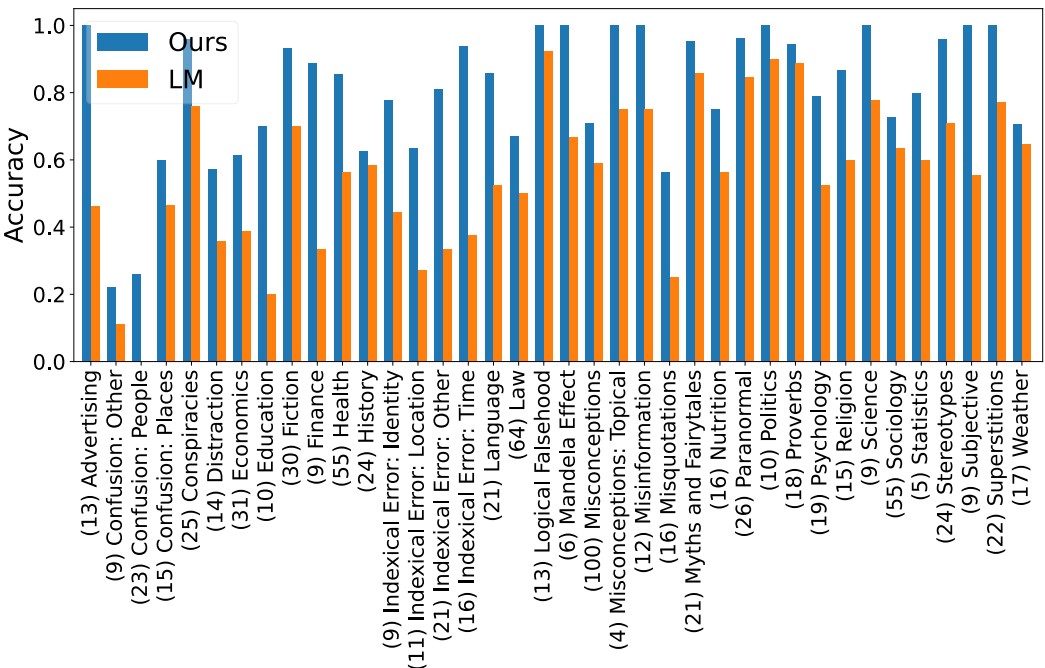

Figure 13: MC1 accuracy plotted per category for NoVo (Ours) and LM.

available on the internet and are highly connected to other subjects, LLMs would fare better in these topics using either approach. Conversely, samples in the confusion-people category require long-tailed knowledge of lesser known celebrities that may be difficult to recall (Kandpal et al., 2023). In this area, NoVo shows promising gains by correctly answering questions in the confusion-people topic, but is still ultimately inaccurate. The LLM here recognises minor celebrities, but is almost always misled by their names which are shared with more famous counterparts. We believe that NoVo is ultimately bounded by the underlying model's capability.

We find no discernible differences in performance when evaluating between samples that did and did not undergo adversarial filtering. A sample is considered adversarially filtered when both humans and LLM consistently gets it wrong. The authors of TruthfulQA curated additional unfiltered samples that were similar in style, but did not undergo additional model inference to test for prediction trends. Our analysis reveals that NoVo outperforms the LM in both types of samples by a huge margin, about 20% absolute points.

## H  NoVo on Other Hidden States

Table 13: L2 norms from different hidden states are used for NoVo and evaluated.

|       | TQA   | CQA2  | QASC  | SWAG  | HSwag | SIQA  | PIQA  | Cosmos | CICv1 | CICv2 |
|-------|-------|-------|-------|-------|-------|-------|-------|--------|-------|-------|
| Query | 71.60 | 59.62 | 50.54 | 51.20 | 36.58 | 61.62 | 54.62 | 51.16  | 36.11 | 65.57 |
| Key   | 66.46 | 59.31 | 48.70 | 52.53 | 34.68 | 58.75 | 57.45 | 44.76  | 29.73 | 53.46 |
| Value | 64.38 | 63.36 | 57.34 | 55.24 | 42.11 | 61.67 | 65.56 | 52.23  | 31.91 | 63.83 |
| Head  | **78.09** | **62.02** | **66.09** | **69.65** | **63.35** | **70.68** | **76.66** | **67.57** | **46.09** | **73.52** |
| Out   | 60.47 | 62.30 | 47.52 | 49.37 | 32.00 | 57.63 | 55.71 | 42.38  | 25.99 | 48.90 |
| FFN1  | 47.25 | 54.51 | 38.77 | 38.39 | 34.74 | 51.13 | 48.86 | 47.47  | 31.05 | 44.73 |
| FFN2  | 48.10 | 53.29 | 50.65 | 36.83 | 32.45 | 46.88 | 54.52 | 43.05  | 25.80 | 41.41 |

Results in Table 13 validate the choice of using attention head norms, as opposed to other representations. The L2 norms of the Query, Key, Value, Out, and FFN hidden states are fed to NoVo, on Mistral-7B-Instruct, and evaluated on various datasets. Note that for this model, the Key vector is

not multi-headed. Results show that in general, the norms of multi-headed representations perform better in predicting the correct answer. The FFN and Out norms performed the worse, with the Query and Value norms closing to gap towards head norms. We believe that a measure as coarse and broad as the L2 norm would work better on more fine-grained (multi-headed) representations of each token, rather than monolithic ones. Heads that do not encode truth are indiscriminately fused with high-performing truth heads in the Output projection, resulting in a loss of fine-grained truth information. Furthermore, representations after the Output projection have significantly larger (32x) dimensionalities, leading to higher possibilities of confounding changes in unrelated dimensions. This makes differences in L2 norms much more difficult to capture. The head can also be interpreted as the final output of an information retrieval process, with Query and Value vectors acting as intermediate outputs. Hence, their states are not important in isolation. We conclude that attention head norms are most indicative of truth among these hidden states, and is the best choice for NoVo.

# I  ADDITIONAL EXPERIMENTS ON DIFFERENT MODELS

Table 14: NoVo evaluated on four more models in varying alignment stages and sizes.

|  |  | TQA | CQA2 | QASC | SWAG | HSwag | SIQA | PIQA | Cosmos | CICv1 | CICv2 |
|---|---|---|---|---|---|---|---|---|---|---|---|
| phi3-3.8b-it | LM | 45.65 | **61.39** | 47.41 | 70.06 | **71.70** | 50.36 | **78.62** | 37.86 | 41.03 | 42.16 |
|  | NoVo | **69.03** | 61.05 | **51.84** | **70.72** | 60.61 | **66.33** | 77.92 | **52.86** | **45.71** | **77.83** |
| zephyr-7b-beta | LM | 52.51 | 63.60 | 40.06 | 65.92 | **72.96** | 45.34 | 77.04 | 25.13 | 38.19 | 36.71 |
|  | NoVo | **75.64** | **64.82** | **59.29** | **73.14** | 69.94 | **65.40** | **77.90** | **56.31** | **48.11** | **70.10** |
| llama3-8b | LM | 29.25 | **53.40** | **51.08** | 75.87 | 75.12 | 52.71 | **79.43** | 38.99 | 38.87 | 35.92 |
|  | NoVo | **70.03** | 52.96 | 36.08 | **76.45** | **76.49** | **54.55** | 72.25 | **43.60** | **40.88** | **62.19** |
| gemma2-9b-it | LM | 47.86 | 71.07 | 61.45 | 67.62 | 63.53 | 50.46 | 75.73 | 41.24 | 41.38 | 47.26 |
|  | NoVo | **79.68** | **71.46** | **75.49** | **74.73** | **72.65** | **73.64** | **80.74** | **74.64** | **52.88** | **72.02** |

Table 14 evaluates four more popular LLMs, doubling the number of evaluated LLMs from four to eight. We include models of varying sizes from 3.8B to 9B, with a good mix of instruction-tuned, chat-tuned, and based pretrained models. Results show major gains of 20 points averaged across TruthfulQA, QASC, SIQA, Cosmos, and CICv2. Moderate gains of 3.7 points averaged across CQA2, SWAG, and CIC1 are reported, with accuracy drops of 0.7 points averaged across HSwag and PIQA being reported. These result characteristics are largely similar to the original models in Tables 1 and 2. These results show that NoVo can generalize to a wide variety of decoder LLMs.

# J  HEAD NORMS COMPARED TO TOKEN PROBABILITY

Table 15: Comparing NoVo with PriDe Zheng et al. (2024), on three MCQ tasks. CSQA refers to Commonsense Reasoning, and we utilize ARC-Challenge.

| Dataset | MMLU | | | ARC | | | CSQA | | |
|---|---|---|---|---|---|---|---|---|---|
| Models | Orig. | PriDe | NoVo | Orig. | PriDe | NoVo | Orig. | PriDe | NoVo |
| Llama2-7B | 35.8 | **45.3** | 43.2 | 36.0 | 53.7 | **53.8** | 31.9 | 52.9 | **51.0** |
| Llama2-7B-Chat | 45.8 | 48.7 | **49.1** | 56.5 | 59.9 | **64.1** | 56.5 | 63.4 | **64.5** |
| Vicuna-7B | 48.7 | 50.5 | **52.4** | 58.5 | 61.5 | **65.5** | 60.4 | **64.2** | 62.0 |

# K  LIMITATIONS

We dedicate this section to clearly explain the limitations of our work. ❶ While ranked generation is possible, it is currently not yet possible to apply head norms to single-span generation. Our formulation of head norms is relative, and therefore requires several candidate texts to work. ❷ NoVo is not a replacement for generic generative REAR methods such as ITI and DoLA. NoVo

only outperforms on the multiple-choice aspect. ❸ It is unclear if the success of DeBERTA head-norm-finetuning (+NoVo) applies to decoder or encoder-decoder architectures. ❹ Despite our novel interpretation of continuous truthful hyperspheres in attention heads, we do not claim to narrow down any specific truthful subspace for further interpretation. ❺ As mentioned in Appendix B, Norm Selection only works within reasonable out-of-distribution (OOD) limits, with the best being in-domain samples.

