# OpenReview forum: "NoVo: Norm Voting off Hallucinations with Attention Heads in Large Language Models"
_ICLR.cc/2025/Conference — ICLR 2025 Poster_

### Official Review · Reviewer_a4Cj · 2024-11-03

**Soundness:** 2
**Presentation:** 3
**Contribution:** 2
**Rating:** 5
**Confidence:** 4

**Summary:**

The authors propose a novel multiple-choice output strategy named NoVo. NoVo identifies correlations between the norms of attention heads at the choice token position and the the accuracy using a small calibration dataset. They leverage this information to make multiple-choice predictions without relying on log-probabilities. The proposed method outperforms various existing hallucination detection methods across different datasets and models. The authors further analyze the voting heads for deeper insights.

**Strengths:**

- The authors did a good job of observing the behavior between attention heads and multiple-choice accuracy.
- The figures are well-designed, and the experiments are comprehensive.
- The hallucination detection task is highly relevant and important today.
- NoVo shows impressive results in multiple-choice benchmarks.

**Weaknesses:**

- This approach has very limited applicability, being useful only for multiple-choice tasks. Therefore, it should be noted that this is not a generic hallucination reduction technique like existing methods such as Truthx.
- It is unclear what makes attention heads special in this context. Similar experiments could potentially be conducted using MLP layers.

- The authors only found a correlation between the norms of the heads and truthfulness. However, it is not explained why such a correlation exists or if there is any causal relationship between the two.
- The direct relationship between the norm of a head and truthfulness is not well-explained. Why should a correlation be expected in the first place? And why use the L2 norm?
- Out-of-distribution experiments are missing. The authors should conduct experiments where NoVo is calibrated on one dataset and tested on another.
- I think the authors should choose between two options. They can either convert their observation about norms and truthfulness into a generic algorithm that works for open-ended generations, or they can deeply explore the reasons behind the observed correlation and write an interpretability paper. Another option might be to reformulate the problem as addressing selection bias or improving multi-choice scenarios, similar to [1].

[1] LARGE LANGUAGE MODELS ARE NOT ROBUST MULTIPLE CHOICE SELECTORS, ICLR 2024

**Questions:**

see Weaknesses.

---

> ### Author Response · Authors · 2024-11-20
> **Official Response to Reviewer a4Cj (1/4)**
>
> **Q1**: This approach has very limited applicability, being useful only for multiple-choice tasks. Therefore, it should be noted that this is not a generic hallucination reduction technique like existing methods such as Truthx.
>
> **A1**: MCQ tasks are not a limitation of our method but rather are used as an evaluation tool to explore and confirm any fundamental internal state misalignments. There are four applications of our solution beyond MCQ tasks:
>
> 1. **Ranked Generation** Our method can be applied beyond MCQs to other task types, such as ranking multiple candidate text spans either generated, or retrieved, by the LLM. Our analysis reveals that head norms spike significantly (up to 83%) in token positions where the proposition is complete (Lines 516-527, 389-402, 430-435, 954-1011). This phenomenon can be used to rank the truthfulness of candidate text spans during open-ended generation.
>
> 2. **In-situ Hallucination Detection** Our findings can be applied beyond MCQs to showcase a more fundamental problem: common and fluent falsehoods are strongly indicative of internal state misalignments. TruthfulQA is a benchmark specifically crafted to mislead LLMs to output common and fluent falsehoods. We show an average 24 point accuracy gain across eight models$^{1}$ in TruthfulQA with head norms, over the language likelihood (Lines 291-298). This difference can be used in hallucination detection on-the-fly, due to its internally encapsulated and lightweight nature, especially for factually incorrect but fluent outputs.
>
> 3. **Adversarial Defence** Our method can be applied beyond MCQs to improve model robustness against textual attacks. An average 6.6 median accuracy gain across four models on all six subtasks of AdversarialGLUE showcase the potential of head norms for building textual adversarial defence (Lines 279-282, 287, 317-323).
>
> 4. **Finetuning** Our method can be applied to improve finetuning accuracy on general tasks beyond MCQs, such as during the alignment stage for better open-ended generations. Experiments on head-norm-finetuning with DeBERTa show an average 0.9 point accuracy gain over standard feature vector fine-tuning, across all nine datasets.
>
> &nbsp;
>
> $^{1}$ The evaluation of four new models have been added, doubling the original number from four to eight.

---

> > ### Comment · Reviewer_a4Cj · 2024-11-22
> >
> > Okay, what I am curious about is whether this method is limited to tasks where we make a selection (or ranking) across possible options. So, this method is not able to decrease the hallucinations in open-end generations as Truthx does, right? If it is not then you should explicitly state this in your paper to not mislead the reader and probably you should compare your method with the methods that focus on multiple choice problems. This would be more fair.
> >
> > I couldn't understand the last claim about fine-tuning which is "Our method can be applied to improve finetuning accuracy on general tasks beyond MCQs, such as during the alignment stage for better open-ended generations." How did you fine-tune DeBERTa and is DeBERTa a generative model here? Can we apply the same idea to a decoder-only model in fine-tuning to reduce hallucinations Can you give more details and provide more explanation, please?

---

> > > ### Author Response · Authors · 2024-11-22
> > > **Official Response to Reviewer a4Cj (1/3)**
> > >
> > > **Q1**: So, this method is not able to decrease the hallucinations in open-end generations as Truthx does, right? If it is not, then you should explicitly state this in your paper to not mislead the reader.
> > >
> > > **A1**: Thank you for the suggestion. We have proposed the possibility of ranked generation in our work, but an important limitation is that it works only when several candidate spans are presented. We have revised our paper to explicitly state this limitation.
> > >
> > > ---
> > > ---
> > >
> > > **Q2**:  you should compare your method with the methods that focus on multiple choice problems. This would be more fair.
> > >
> > > **A2**: Thank you for your suggestions. TruthX and other REAR methods are the current state-of-the-art in TruthfulQA MC1 accuracy [1]. This makes for a fair and strong comparison. Additionally, we are happy to present comparisons to [2]. We have also revised our paper to explicitly convey to readers that our method outperforms TruthX on the multiple-choice aspect only.
> > >
> > > | Dataset |       |  MMLU |      |       |  ARC  |      |       |  CSQA |      |
> > > |----------------|:-----:|:-----:|:----:|:-----:|:-----:|:----:|:-----:|:-----:|:----:|
> > > | Models | Orig. | PriDe | NoVo | Orig. | PriDe | NoVo | Orig. | PriDe | NoVo |
> > > | Llama2-7B|  35.8 |  45.3 | 43.2 |  36.0 |  53.7 | 53.8 |  31.9 |  52.9 | 51.0 |
> > > | Llama2-7B-Chat |  45.8 |  48.7 | 49.1 |  56.5 |  59.9 | 64.1 |  56.5 |  63.4 | 64.5 |
> > > | Vicuna-7B|  48.7 |  50.5 | 52.4 |  58.5 |  61.5 | 65.5 |  60.4 |  64.2 | 62.0 |
> > >
> > > ---
> > > ---
> > >
> > > **Q3**: How did you fine-tune DeBERTa and is DeBERTa a generative model here? Can we apply the same idea to a decoder-only model in fine-tuning to reduce hallucinations Can you give more details and provide more explanation, please?
> > >
> > > **A3**: We are happy to give more details and explain.
> > > 1. **Is DeBERTa a generative model here?** DeBERTa is not generative, it is an encoder LLM.
> > > 2. **How did you fine-tune DeBERTa?** We fine-tune DeBERTa by packing all head norm scalars into a vector, and feeding only that vector to the final classifier head. Our experiments show this approach outperforming standard finetuning, which instead uses the last feature vector for classification.
> > > 3. **Can we apply the same idea to a decoder-only model in fine-tuning to reduce hallucinations** Yes, this can be also done for decoder LLMs in the exact same manner: feed all head norms to the classifier head. Before RLHF alignment, decoder models usually undergo supervised finetuning on human-preferred pairs of prompt-responses. Finetuning plays an important role in alignment. Alignment plays an important role in reducing hallucinations during generation. We have shown that head norms improve finetuning performance in DeBERTa but not decoder LLMs. Future works can work on specific alignment datasets.
> > >
> > > We have revised the paper to clarify all three points, including the portion on alignment as future works.
> > >
> > > [1] https://paperswithcode.com/sota/question-answering-on-truthfulqa
> > > [2] Large Language Models Are Not Robust Multiple Choice Selectors

---

> ### Author Response · Authors · 2024-11-20
> **Official Response to Reviewer a4Cj (2/4)**
>
> **Q2**: It is unclear what makes attention heads special in this context. Similar experiments could potentially be conducted using MLP layers.
>
> **A2**: Thank you for your suggestion. We are happy to present our experiments on both MLP (FFN) and QKVO states. The results show an average 23.9 point loss in accuracy when using MLP norms. This is improved to an average 12.5 point loss when using multi-headed states Q and V. Head norms outperform all representations in all datasets.
>
> There are two reasons why attention head norms are good in this context.
>
> 1. **Multi-Heads are Fine-Grained** A measure as coarse and broad as the L2 norm would work much better on more fine-grained (multi-headed) representations of each token, rather than monolithic ones. This is supported by experimental results, which shows lower accuracies for representations that are not split into multiple heads (K, O, FFN1, and FFN2), with FFN and O vector norms having the lowest scores.
>
> 2. **Heads are the Final Representation of Information** The head is the final output of the information retrieval process described by the QKV operation. Here, Q and V are intermediate outputs used to retrieve relevant information; their states are not important in isolation. This is supported by experimental results, which shows better accuracies due to their multi-headed nature, but still lower than the head.
>
> |          |    tqa    |   csqa2   |    qasc   |    swag   |   hswag   |    siqa   |    piqa   |   cosmo   |   cicv1   |   cicv2   |
> |:--------:|:---------:|:---------:|:---------:|:---------:|:---------:|:---------:|:---------:|:---------:|:---------:|:---------:|
> | Query    |   71.60   |   59.62   |   50.54   |   51.20   |   36.58   |   61.62   |   54.62   |   51.16   |   36.11   |   65.57   |
> | Key$^{1}$      |   66.46   |   59.31   |   48.70   |   52.53   |   34.68   |   58.75   |   57.45   |   44.76   |   29.73   |   53.46   |
> | Value    |   64.38   |   63.36   |   57.34   |   55.24   |   42.11   |   61.67   |   65.56   |   52.23   |   31.91   |   63.83   |
> | **Head** | **78.09** | **62.02** | **66.09** | **69.65** | **63.35** | **70.68** | **76.66** | **67.57** | **46.09** | **73.52** |
> | Out      |   60.47   |   62.30   |   47.52   |   49.37   |   32.00   |   57.63   |   55.71   |   42.38   |   25.99   |   48.90   |
> | FFN1     |   47.25   |   54.51   |   38.77   |   38.39   |   34.74   |   51.13   |   48.86   |   47.47   |   31.05   |   44.73   |
> | FFN2     |   48.10   |   53.29   |   50.65   |   36.83   |   32.45   |   46.88   |   54.52   |   43.05   |   25.80   |   41.41   |
>
> &nbsp;
>
> $^{1}$ Experiments are conducted on Mistral-7b-Instruct, which does not use multi-headed K vectors.
>
> ---
> ---
> **Q3**: The authors only found a correlation between the norms of the heads and truthfulness. However, it is not explained why such a correlation exists or if there is any causal relationship between the two.
>
> **A3**: There are three reasons why such a correlation could exist:
>
> 1. **Proposition Completion** We find evidence that head norms are correlated with truth because of the completion of factual propositions. We see that head norms spike significantly in token positions where the proposition is complete (Lines 389-402, 430-435, 954-1011). Taking the head norms prematurely before completion results in catastrophic loss of performance (Lines 516-527).
>
> 2. **Factual Association** We find evidence that head norms are correlated with truth because of the attention towards specific factual associations, such as between ‘pigs’ and ‘fly’. We see that head norms spike significantly in token positions where these associations occur (Lines 389-402, 416,420, 954-1011). Perturbing these associations with character insertions causes catastrophic loss of performance, and vice versa (Line 522).
>
> 3. **Self-organisation in High-Dimensional Space** Representations are found to self-organise in human-meaningful clusters, with mnist digits clustering based on stroke similarity [1], and word representations clustering based on human constructs such as gender [2]. It is plausible that truthful statements are self-organised as well. Small semantic shifts in truth can lead to large but organised movements in high-dimensional space [3], causing points to lay in a predictably different hypersphere surface. This mechanism is a possible reason why truth and L2 head norms are correlated.
>
> We will supplement these analyses in the revised version.
> \
> [1] Self-classifying MNIST Digits, accessed online at https://distill.pub/2020/selforg/mnist/.
> [2] Efficient Estimation of Word Representations in Vector Space, Arxiv 2013.
> [3] Semantics in High-Dimensional Space, Frontiers 2021.

---

> > ### Comment · Reviewer_a4Cj · 2024-11-22
> >
> > It's really interesting that when you do the same analysis with Out there are huge performance drops although there is a linear relationship between Out and Head. What could possibly happen in a linear layer that would affect the factuality so much? Actually, my point is that the connection between the head output norm and improved accuracy performance doesn't really tell a convincing story here in terms of interpretability. Also, the correlation you show is not a direct correlation between the norm of heads and the factuality. You first process the norms of all heads in a non-linear way then you use labeled data to find heads that are gonna be used in voting. This is actually a training and you use all heads in your training. From the interpretability perspective, what is the take here? Should we say the factuality information is encoded in heads so we can extract this information with our training strategy? Well, yes of course that information is encoded in the heads, that is a trivial conclusion. Can you imagine the opposite is true that none of the heads have any encoded information about the factuality of a generation? I think your novelty here is to propose an efficient training-like strategy to get this information from heads by using a few labels and using it in multiple-choice tasks. This is also valuable but your narrative should follow this story and your benchmark should include other algorithms that focus on improving multi-choice performance of LLMs not generic hallucination reduction methods like Truthx.

---

> > > ### Author Response · Authors · 2024-11-22
> > > **Official Response to Reviewer a4Cj (2/3)**
> > >
> > > **Q4**: It's really interesting that when you do the same analysis with Out there are huge performance drops although there is a linear relationship between Out and Head. What could possibly happen in a linear layer that would affect the factuality so much?
> > >
> > > **A4**: There are two reasons why the Out linear layer causes a huge (but linear) performance drop over the Head:
> > > 1. Heads that do not encode truth are indiscriminately fused with high-performing truth heads. This results in fine-grained information about truth being lost.
> > > 2. The dimensionality of the Out vector is significantly larger (32x larger),  with higher possibilities of confounding changes in unrelated dimensions. This makes differences in L2 norms much more difficult to capture in Out.
> > >
> > > We have revised our paper to add this analysis inside.
> > >
> > > ---
> > > ---
> > > **Q5**: The correlation you show is not a direct correlation between the norm of heads and the factuality. You first process the norms of all heads in a non-linear way then you use labeled data to find heads that are gonna be used in voting. This is actually a training and you use all heads in your training.
> > >
> > > **A5**: KIndly allow us to clarify: If there was no direct correlation between head norms and truth, then no amount of processing will help.
> > >
> > > We stress that **individual head norms natively increase (or decrease) reliably when presented with various true-false statements, without any processing**. The norm changes from just a single mistral-7b head can already achieve an impressive 69.77% on TruthfulQA-MC1, 9 points above GPT4. NoVo _uses_ this underlying direct correlation, rather than modifying it. First, this “zeroth-order training” you mentioned is simply a filtering operation to remove low-correlation heads whose predictions are still above random, but will hurt the ensemble. Then, we ensemble the predictions of high-correlating heads to boost accuracy. These two processing steps will not make sense if there is no native correlation between the norms of heads and factuality to begin with. **Our experiments show a direct correlation between head norms and factuality.**
> > >
> > > _Table 1: Individual head predictions on TruthfulQA without any processing or ‘training’. Note that >75% of heads perform above random baseline, indicating a non-trivial direct correlation with truth_
> > >
> > > | Count |  Mean |  Std |  Min  |  25Q  |  50Q  |  75Q  |  Max  |
> > > |:-----:|:-----:|:----:|:-----:|:-----:|:-----:|:-----:|:-----:|
> > > |  1024 | 37.29 | 8.00 | 20.32 | 31.43 | 35.86 | 41.49 | 69.77 |
> > > ---
> > > ---
> > > **Q6**: From the interpretability perspective, what is the take here?
> > >
> > > **A6**: Our interpretability take is **Continuous Variations in Hypersphere Surfaces**. While previous works (ITI, CCS) reveal binary true-false clusters, we show that the head representations of sequences lay on different hypersphere surfaces (L2 norm) [1], depending on how their truth content varies. A less truthful sequence will lay on a smaller hypersphere. This continuous interpretation is not shown in the binary clusters of previous works, whose point-to-point distances did not reflect varying truth [2]. Our insight, combined with previous works, paints a clearer intuition in how head representations self-organise around truth. This opens exciting interpretability questions: can a curated dataset isolate a defined subspace in heads where these hypersphere (L2 norm) variations can be projected onto [3], by empirically probing for meaningful skills (other than factuality).
> > >
> > > [1] Debarre, Olivier. Higher-dimensional algebraic geometry. Vol. 3. New York: Springer, 2001.
> > > [2] Inference-Time Intervention: Eliciting Truthful Answers from a Language Model.
> > > [3] Deep subspace clustering networks. NeuRIPS 2017.
> > >
> > > ---
> > > ---
> > >
> > > **Q7**: I think your novelty here is to propose an efficient training-like strategy to get this information from heads by using a few labels and using it in multiple-choice tasks. This is also valuable but your narrative should follow this story and your benchmark should include other algorithms that focus on improving multi-choice performance of LLMs not generic hallucination reduction methods like Truthx.
> > >
> > > **A7**: We have revised sentences in our paper to convey your suggested contributions for improving multiple-choice scenarios. Following your advice, we also revised our paper to explicitly clarify that it is not a generic hallucination reduction method. In response to your feedback, we have added a new multi-choice benchmark in our paper that is not a generic hallucination reduction method.

---

> ### Author Response · Authors · 2024-11-20
> **Official Response to Reviewer a4Cj (3/4)**
>
> **Q4**: The direct relationship between the norm of a head and truthfulness is not well-explained. Why should a correlation be expected in the first place? And why use the L2 norm?
>
> **A4**: We list three reasons why a correlation was expected, and why to use the L2 norm.
>
> 1. **Evidence of Correlation in Prior Works** Previous studies by Burns et al. [1], Azaria et al. [2], Zou et al. [3], Li et al. [4], and Chen et al. [5] demonstrated that the latent representations of LLMs can be linearly classified into true-false clusters. While many of these works focused on the MLP, Li et al. extended this to individual attention heads, confirming that certain heads encode truth-related features. These findings strongly suggest that some LLM hidden states self-organise around truthfulness, motivating us to explore whether L2 norms of specific heads similarly reflect this pattern.
>
> 2. **Use of L2 Norms in Other Domains** In computer vision, studies have shown that the L2 norm of feature vectors in the final layer of CNNs correlates with perceived image quality [6,7]. Poor-quality images result in weaker, sparser vector representations [8]. Drawing from this analogy, we hypothesised that intermediate L2 norms in transformers might similarly capture the strength of features relevant to "truthfulness" in language models, even across layers and architectures.
>
> 3. **Conceptual Basis of Representation Learning** Transformers encode diverse language features in their intermediate representations, which often self-organise along meaningful dimensions [9-11]. It is plausible that one of these dimensions reflects the alignment of propositions with reality (i.e., truthfulness) [12]. For instance, features related to coherent concepts (e.g., "plane - passenger - ticket") might express stronger magnitudes in these representations, leading to higher (or, if negatively correlated, lower) L2 norms in certain attention heads.
> &nbsp;
>
> [1] Discovering Latent Knowledge in Language Models Without Supervision, ICLR 2023
> [2] The Internal State of an LLM Knows When It’s Lying, EMNLP 2023
> [3] Representation Engineering: A Top-Down Approach to AI Transparency, Arxiv 2023
> [4] Inference-Time Intervention: Eliciting Truthful Answers from a Language Model, NeurIPS 2023
> [5] Truth Forest: Toward Multi-Scale Truthfulness in Large Language Models through Intervention without Tuning, AAAI 2024.
> [6] Deep Learning of Human Visual Sensitivity in Image Quality Assessment Framework, CVPR 2017
> [7] Deep Objective Quality Assessment Driven Single Image Super-Resolution, IEEE-TMM 2019
> [8] Wang, Z., & Bovik, A. C. (2006). Modern Image Quality Assessment. In Synthesis lectures on image, video, and multimedia processing. https://doi.org/10.1007/978-3-031-02238-8.
> [9] Girolami, M. (1999). Self-Organising Neural Networks. In Perspectives in neural computing. https://doi.org/10.1007/978-1-4471-0825-2
> [10] Li, Ping. "Language acquisition in a self-organising neural network model." Connectionist Models of Development. Psychology Press, 2004. 112-142.
> [11] Self-classifying MNIST Digits, accessed online at https://distill.pub/2020/selforg/mnist/
> [12] Efficient Estimation of Word Representations in Vector Space, Arxiv 2013

---

> ### Author Response · Authors · 2024-11-20
> **Official Response to Reviewer a4Cj (4/4)**
>
> **Q5**: Out-of-distribution experiments are missing. The authors should conduct experiments where NoVo is calibrated on one dataset and tested on another.
>
> **A5**: We conduct out-of-distribution (OOD) experiments on NoVo-Mistral-7B-Instruct. Here, NoVo is calibrated on one dataset and tested on another. We calibrate on ten datasets: Arc-Easy, CQA2, QASC, SWAG, HSwag, SIQA, PIQA, Cosmos, CICv1, and CICv2 in a pairwise manner, and record the standard deviations of accuracies. The standard deviation reflects the spread of the OOD accuracy of a dataset when calibrated on various other datasets.
> &nbsp;
> |               | Arc Easy | cqa2 | qasc | swag | hswag | siqa | piqa | cosmos | cicv1 | cicv2 |
> |---------------|:--------:|:----:|:----:|:----:|:-----:|:----:|:----:|:------:|:-----:|:-----:|
> | Accuracy Std. |   2.76   | 0.59 | 4.66 | 6.84 |  9.76 | 2.12 | 3.37 |  5.13  |  3.79 |  3.85 |
>  &nbsp;
>
> The accuracy standard deviation show that OOD performance is mostly dependent on the evaluated dataset, rather than our method. For example, our method can be calibrated significantly OOD on multi-turn CICv1 dialogues, sentence completion SWAG, or factual-style Arc-Easy, and evaluated for CQA2 commonsense yes-no questions with only a 0.59 standard deviation. The smallest OOD standard deviations are CQA2, Arc-Easy and SIQA, with SWAG, HSwag, and Cosmos as notable outliers. We note that we specifically chose to maximise the OOD difference between each dataset, by manually checking that question topics and formats are highly varied, to ensure that our method can truly generalise. We also note that the SOTA performance achieved by NoVo on TruthfulQA was calibrated OOD on Arc-Easy.
>
> ----
> ----
> **Q6**: I think the authors should choose between two options. They can either convert their observation about norms and truthfulness into a generic algorithm that works for open-ended generations, or they can deeply explore the reasons behind the observed correlation and write an interpretability paper. Another option might be to reformulate the problem as addressing selection bias or improving multi-choice scenarios, similar to [1].
>
> **A6**: We are happy to note that our paper’s contributions are well-aligned with your suggestion. NoVo addresses the selection bias and improves multi-choice scenarios (Lines 85-87). NoVo enables LLMs to do better in multi-choice scenarios via two mechanisms: First NoVo circumvents the selection bias problem identified in [1] by operating on entire text spans, rather than letter options. Second NoVo improves robustness to common falsehoods, by avoiding the popular use of log likelihoods in multi-choice scenarios (Line 55). If we placed our paper side-by-side with [1]:
>
> **Zheng et al. ICLR 2024 [1]**
> 1. Viewed poor MCQ performance through the lens of selection bias.
> 2. Formulated the selection bias problem as an a priori probabilistic problem.
> 3. Proposed PriDe to separate the prior and prediction distribution, to improve MCQ, resulting in ~1.5% improvement in 2 datasets.
> 4. Provides an interpretive exploration of the selection bias problem.
>
> **Our Paper**
> 1. View poor MCQ performance through the lens of hallucinations.
> 2. Formulated the hallucination problem as an internal state misalignment problem.
> 3. Proposed NoVo to operate directly on internal states (head norms) to improve MCQ, resulting in ~10% improvement across 21 datasets.
> 4. Provides an interpretive exploration of the internal misalignment problem.
>
> Zheng et al. contributed to the selection bias problem using MCQ tasks to demonstrate their hypothesis. We contributed to the internal-state misalignment problem using MCQ tasks to demonstrate our hypothesis.
>
> We will clarify this concern in the revision of our related works section.
>
> [1] Large Language Models Are Not Robust Multiple Choice Selectors, ICLR 2024

---

> ### Comment · Reviewer_a4Cj · 2024-11-22
>
> May you also share the full table of results instead of only stds?
>
> It would be really good if you included [1] in your benchmark.
>
>
> I will increase my score to 5 but I am still on the rejection side (you can think my score is 4 but there is no such an option) unless other reviewers and/or authors convince me. I appreciate the efforts of the authors and the detailed experimental results in their paper. I will be happier if the authors bravely discuss all the limitations of their proposed algorithm in the paper. A good paper doesn't have to solve all problems in an area. A paper showing its limitations with comprehensive benchmarks is what our community needs. I know we feel pressure to make big improvements in the paper to get it accepted that's why we tend to overclaim in the papers but this only hurts the scientific progress.

---

> ### Author Response · Authors · 2024-11-22
> **Official Response to Reviewer a4Cj (3/3)**
>
> **Q8**: May you also share the full table of results instead of only stds?
>
> **A8**: We are happy to present the full OOD table replicated from the paper. We note that calibration (i.e. zeroth order training) works within reasonable OOD limits (i.e. TruthfulQA is calibrated on Arc-Easy) and requires very little data. There are obvious limitations when calibrating to a significantly OOD dataset (multi-turn dialogue CICv1 to ambiguity-focused PIQA). This limitation has been clarified in the paper.
>
> |        |  ArcE |  CQA2 |  QASC |  SWAG | HSwag |  SIQA |  PIQA | Cosmos | CICv1 | CICv2 |
> |:------:|:-----:|:-----:|:-----:|:-----:|:-----:|:-----:|:-----:|:------:|:-----:|:-----:|
> | ArcE   | 84.70 | 59.94 | 55.94 | 59.66 | 49.64 | 70.98 | 75.14 |  53.70 | 37.96 | 70.74 |
> | CQA2   | 75.39 | 61.67 | 56.80 | 54.55 | 41.86 | 66.33 | 67.46 |  63.15 | 39.98 | 65.25 |
> | QASC   | 80.93 | 60.80 | 67.60 | 61.23 | 50.24 | 68.99 | 73.56 |  66.83 | 40.69 | 65.57 |
> | SWAG   | 80.93 | 59.86 | 59.61 | 74.19 | 64.63 | 69.55 | 76.17 |  57.96 | 39.41 | 66.32 |
> | HSwag  | 80.04 | 60.61 | 57.24 | 72.83 | 70.11 | 65.15 | 76.50 |  58.32 | 41.33 | 63.65 |
> | SIQA   | 83.81 | 59.74 | 60.15 | 58.89 | 43.63 | 70.21 | 71.98 |  57.69 | 39.83 | 72.24 |
> | PIQA   | 82.26 | 59.90 | 54.64 | 67.38 | 61.69 | 70.93 | 79.22 |  54.24 | 40.54 | 70.63 |
> | Cosmos | 77.38 | 60.02 | 57.13 | 60.33 | 48.51 | 68.47 | 71.87 |  68.94 | 42.85 | 70.31 |
> | CICv1  | 81.15 | 60.13 | 58.86 | 63.84 | 54.26 | 67.55 | 74.59 |  62.78 | 51.48 | 72.02 |
> | CICv2  | 79.82 | 59.98 | 49.14 | 54.59 | 42.86 | 65.76 | 70.57 |  57.25 | 43.89 | 75.73 |
>
> ---
> ---
>
> **Q9**: It would be really good if you included [1] in your benchmark.
>
> **A9**: We have included [1] in our evaluations in a new revision of our paper. We reproduce the table here for your convenience.
>
> | Dataset |       |  MMLU |      |       |  ARC  |      |       |  CSQA |      |
> |----------------|:-----:|:-----:|:----:|:-----:|:-----:|:----:|:-----:|:-----:|:----:|
> | Models | Orig. | PriDe | NoVo | Orig. | PriDe | NoVo | Orig. | PriDe | NoVo |
> | Llama2-7B|  35.8 |  45.3 | 43.2 |  36.0 |  53.7 | 53.8 |  31.9 |  52.9 | 51.0 |
> | Llama2-7B-Chat |  45.8 |  48.7 | 49.1 |  56.5 |  59.9 | 64.1 |  56.5 |  63.4 | 64.5 |
> | Vicuna-7B|  48.7 |  50.5 | 52.4 |  58.5 |  61.5 | 65.5 |  60.4 |  64.2 | 62.0 |
>
> ---
> ---
>
> **Q10**: I will be happier if the authors bravely discuss all the limitations of their proposed algorithm in the paper.
>
> **A10**: We have added a limitation section in the paper per your suggestion:
>
> We dedicate this section to clearly explain the limitations of our work. **(1)** While ranked generation is possible, it is currently not yet possible to apply head norms to single-span generation. Our formulation of head norms is relative, and therefore requires several candidate texts to work. **(2)** NoVo is not a replacement for generic generative REAR methods such as ITI and DoLA. NoVo only outperforms on the multiple-choice aspect. **(3)** It is unclear if the success of DeBERTA head-norm-finetuning (+NoVo) applies to decoder or encoder-decoder architectures. **(4)** Despite our novel interpretation of continuous truthful hyperspheres in attention heads, we do not claim to narrow down any specific truthful subspace for further interpretation. **(5)** As mentioned in Appendix B, Norm Selection only works within reasonable out-of-distribution (OOD) limits, with the best being in-domain samples.

---

> ### Author Response · Authors · 2024-11-22
> **Revision Summary for Reviewer a4Cj**
>
> Dear Reviewer a4Cj,
>
> The uploaded PDF will highlight all changes from the initial version, so kindly allow us to summarise the second-round revision of the paper here, based on your second-round feedback.
>
> Line Number: Comment
> `053`: explicit clarification that our method is for multi-choice scenarios.
> `086`: explicitly stated that ranked generation is for future works.
> `098`: clarified REAR methods that are generic hallucination mitigation techniques, to avoid confusing readers.
> `110`: clarified REAR methods that are generic hallucination mitigation techniques, to avoid confusing readers.
> `113`: explicit clarification that our method is for multi-choice scenarios.
> `377`: explicitly stated that ranked generation is for future works.
> `380`: explicitly stated that finetuning on decoder models for alignment is for future works.
> `1138`: Explained why Out norms factuality performs falters when compared to head.
> `1179`: Included evaluation for [Large Language Models Are Not Robust Multiple Choice Selectors].
> `1183`: Included a section detailing limitations of our work.

---

> > ### Comment · Reviewer_a4Cj · 2024-11-25
> >
> > I thank the reviewers for the explanation. I think we are on the same page right now. I invite other reviewers to read our discussion thread in detail.

---

### Official Review · Reviewer_GDLP · 2024-11-04

**Soundness:** 3
**Presentation:** 4
**Contribution:** 3
**Rating:** 8
**Confidence:** 3

**Summary:**

* The authors propose a simple, novel technique to improve LLM factual accuracy. They first find truth-correlated attention head norms, then ensemble these with majority voting at inference time to choose answers to MC questions.
* The authors conduct comprehensive experiments on a diverse range of datasets and use several different models. On many datasets they find massive effects from their intervention, improving on sota by 20+ percentage points in some cases.

**Strengths:**

* Empirical results are strong and consistent across models, with substantial improvements over previous methods
* The method is simple and cheap, requiring no specialized training or tools.
* Experimental evaluation is comprehensive, testing across 20 diverse datasets. The authors also evaluate generalizability through finetuning and adversarial robustness tests
* Good error analysis that helps understand the method's capabilities and limitations, including detailed investigation of how different voter types contribute and where/why the method fails

**Weaknesses:**

* While the method is simple, I find the conceptual motivation unclear. The authors posit that for some heads, the L2 norm is correlated to the truthfulness of a sequence. Why? What makes this a reasonable thing to expect, conceptually?
* These experiments are done exclusively on multiple-choice QA datasets, and it seems that it would not be possible to use this method to reduce hallucination in open-ended generation.
* Evaluating on a wider range of models would help provide more confidence in the method, especially testing on currently popular models like Llama 3.

**Questions:**

* In Table 2, very different results are obtained for CICv1 and CICv2. Is there any explanation for why the results differ so much here, or in general why the results vary so much across datasets?
* What is the variability of results when using different sets of randomly drawn samples for norm selection?

---

> ### Author Response · Authors · 2024-11-20
> **Official Response to Reviewer GDLP (1/3)**
>
> **Q1**: While the method is simple, I find the conceptual motivation unclear. The authors posit that for some heads, the L2 norm is correlated to the truthfulness of a sequence. Why? What makes this a reasonable thing to expect, conceptually?
>
> **A1**: We list three reasons why this was a reasonable thing to expect, conceptually.
>
> 1. **Evidence from Prior Works** Previous studies by Burns et al. [1], Azaria et al. [2], Zou et al. [3], Li et al. [4], and Chen et al. [5] demonstrated that the latent representations of LLMs can be linearly classified into true-false clusters. While many of these works focused on FFN outputs, Li et al. extended this to individual attention heads, confirming that certain heads encode truth-related features. All these findings strongly suggest that some LLM hidden states self-organise around truthfulness, motivating us to explore whether L2 norms of specific heads similarly reflect this pattern.
>
> 2. **Insights from L2 Norms in Other Domains** In computer vision, studies have shown that the L2 norm of feature vectors in the final layer of CNNs correlates with perceived image quality [6,7]. Poor-quality images result in weaker, sparser vector representations [8]. Drawing from this analogy, we hypothesised that intermediate L2 norms in transformers might similarly capture the strength of features relevant to "truthfulness" in language models, even across layers and architectures.
>
> 3. **Conceptual Basis in Representation Learning** Transformers encode diverse language features in their intermediate representations, which often self-organise along meaningful dimensions [9-11]. It is plausible that one of these dimensions reflects the alignment of propositions with reality (i.e., truthfulness) [12]. For instance, features related to coherent concepts (e.g., "plane - passenger - ticket") might express stronger magnitudes in these representations, leading to higher (or, if negatively correlated, lower) L2 norms in certain attention heads.
>
> &nbsp;
>
> [1] Discovering Latent Knowledge in Language Models Without Supervision, ICLR 2023
> [2] The Internal State of an LLM Knows When It’s Lying, EMNLP 2023
> [3] Representation Engineering: A Top-Down Approach to AI Transparency, Arxiv 2023
> [4] Inference-Time Intervention: Eliciting Truthful Answers from a Language Model, NeurIPS 2023
> [5] Truth Forest: Toward Multi-Scale Truthfulness in Large Language Models through Intervention without Tuning, AAAI 2024.
> [6] Deep Learning of Human Visual Sensitivity in Image Quality Assessment Framework, CVPR 2017
> [7] Deep Objective Quality Assessment Driven Single Image Super-Resolution, IEEE-TMM 2019
> [8] Wang, Z., & Bovik, A. C. (2006). Modern Image Quality Assessment. In Synthesis lectures on image, video, and multimedia processing. https://doi.org/10.1007/978-3-031-02238-8.
> [9] Girolami, M. (1999). Self-Organising Neural Networks. In Perspectives in neural computing. https://doi.org/10.1007/978-1-4471-0825-2
> [10] Li, Ping. "Language acquisition in a self-organising neural network model." Connectionist Models of Development. Psychology Press, 2004. 112-142.
> [11] Self-classifying MNIST Digits, accessed online at https://distill.pub/2020/selforg/mnist/
> [12] Efficient Estimation of Word Representations in Vector Space, Arxiv 2013

---

> ### Author Response · Authors · 2024-11-20
> **Official Response to Reviewer GDLP (2/3)**
>
> **Q2**: These experiments are done exclusively on multiple-choice QA datasets, and it seems that it would not be possible to use this method to reduce hallucination in open-ended generation.
>
> **A2**: Our method can be reformulated to adapt to open-ended generation. During generation, the LLM can either decode or externally retrieve multiple candidate text spans. Our analysis reveals that head norms spike significantly by up to 83% when compared to a different candidate text, in token positions where the proposition is complete (Lines 516-527, 389-402, 430-435, 954-1011). This phenomenon can be used to rank the truthfulness of candidate text spans during open-ended generation.
>
> In addition, experiments on AdversarialGLUE where our method achieves an average 6.6 median accuracy gain across all models on all 6 subtasks, and head-norm-finetuning on DeBERTa, where our method achieves an average 0.9 point gain across all models on all 9 datasets, show that our method can play a key role outside of MCQ tasks: improving adversarial defence and finetuning accuracy during alignment, for more desirable open-ended generations.
>
> ---
> ---
>
> **Q3**: Evaluating on a wider range of models would help provide more confidence in the method, especially testing on currently popular models like Llama 3.
>
> **A3**:  Thank you for your suggestion. We are happy to present additional evaluations on four more currently popular models, doubling the number of evaluated models from 4 to 8. We included varying sizes ranging from 3.8b to 9b, and ensured a good mix of instruction-tuned, chat-tuned, and based pretrained models. Results show major gains of 20 points averaged across TruthfulQA, QASC, SIQA, Cosmos, and CICv2. Moderate gains of 3.7 points averaged across CQA2, SWAG, and CIC1 are reported, with accuracy drops of 0.7 points averaged across HSwag and PIQA being reported. We note that these performance characteristics are largely similar to the original models.
> |                |      |    tqa    |   csqa2   |    qasc   |    swag   |   hswag   |    siqa   |    piqa   |    cosm   |    cic1   |    cic2   |
> |----------------|------|:---------:|:---------:|:---------:|:---------:|:---------:|:---------:|:---------:|:---------:|:---------:|:---------:|
> | phi3-3.8b-it   | lm   |   45.65   | **61.39** |   47.41   |   70.06   | **71.70** |   50.36   | **78.62** |   37.86   |   41.03   |   42.16   |
> |                | novo | **69.03** |   61.05   | **51.84** | **70.72** |   60.61   | **66.33** |   77.92   | **52.86** | **45.71** | **77.83** |
> | zephyr-7b-beta | lm   |   52.51   |   63.60   |   40.06   |   65.92   | **72.96** |   45.34   |   77.04   |   25.13   |   38.19   |   36.71   |
> |                | novo | **75.64** | **64.82** | **59.29** | **73.14** |   69.94   | **65.40** | **77.90** | **56.31** | **48.11** | **70.10** |
> | llama3-8b      | lm   |   29.25   | **53.40** | **51.08** |   75.87   |   75.12   |   52.71   | **79.43** |   38.99   |   38.87   |   35.92   |
> |                | novo | **70.03** |   52.96   |   36.08   | **76.45** | **76.49** | **54.55** |   72.25   | **43.60** | **40.88** | **62.19** |
> | gemma2-9b-it   | lm   |   47.86   |   71.07   |   61.45   |   67.62   |   63.53   |   50.46   |   75.73   |   41.24   |   41.38   |   47.26   |
> |                | novo |   79.68   | **71.46** | **75.49** | **74.73** | **72.65** | **73.64** | **80.74** | **74.64** | **52.88** | **72.02** |

---

> ### Author Response · Authors · 2024-11-20
> **Official Response to Reviewer GDLP (3/3)**
>
> **Q4**: In Table 2, very different results are obtained for CICv1 and CICv2. Is there any explanation for why the results differ so much here, or in general why the results vary so much across datasets?
>
> **A4**:  The reason why results vary across datasets can be attributed to three types of questions:
>
> 1. **Questions with Strong Stereotype Assumptions** The first type of question requires strong stereotypes to disambiguate equally likely answers (Section 4.3). We find that our method avoids assuming strong stereotypes, which can be helpful in ambiguous questions. For example, in PIQA, when asked whether a book or milk should be placed on the shelf. Both are correct, but the strong assumption here (and ground truth answer) is ‘book’. A larger proportion of these questions can hurt results.
>
> 2. **Questions with Identical Answers** The second type of questions admits nearly identical answers, for example: choosing between “no”, and “no, he did not”. For these questions, the correct answer can vary widely, which causes unpredictable result fluctuations.
>
> 3. **Questions with Misleading Phrasing** The third type of questions contains misleading phrasings and tries to elicit fluent falsehoods, such as is mostly found in TruthfulQA. Our method avoids assuming strong stereotypes and hence performs well for these types of questions. A larger proportion of these questions can improve results.
> ---
> ---
> **Q5**: What is the variability of results when using different sets of randomly drawn samples for norm selection?
>
> **A5**: Random variations experiments are conducted over 200 runs, across all ten datasets (TruthfulQA, CQA2, QASC, SWAG, HSwag, SIQA, PIQA, Cosmos, CICv1, and CICv2) and across all four models (10 x 4 = 40 reports, each 200 runs). Standard deviations are all within 1.5 points, with the exception of Llama2-7b-Cosmos at 1.64, and Vicuna-7b-QASC at 1.53. Interquartile ranges are all within 2.3 points. **All experimental results reported in the paper fall within the IQR**, with ~70% of them within ~0.5 points from the median. Our random variation experiments show that there is **no over-reporting of results**. As the full experiment table is quite large, we kindly direct the reviewer to Appendix D for more details.

---

> ### Author Response · Authors · 2024-11-25
> **Gentle Reminder: Period Ending – We Anticipate Your Feedback!**
>
> Dear Reviewer GDLP,
>
> Thank you again for your comments, effort and time spent reviewing this paper.
>
> We gently remind the reviewer that the discussion stage is ending. We deeply appreciate your valuable insights in helping us improve our manuscript's quality. Once this stage ends, we will be unable to respond.
>
> Kindly let us know if you have further comments about this paper. We look forward to your suggestions!
>
> Regards,
> Authors of Submission 3392

---

### Official Review · Reviewer_3eKd · 2024-11-06

**Soundness:** 3
**Presentation:** 4
**Contribution:** 2
**Rating:** 5
**Confidence:** 4

**Summary:**

The paper introduces Norm Voting (NoVo), a lightweight method designed to reduce hallucinations in LLMs using attention head norms.

- Norm Voting automatically selects attention head norms that correlate with truth using a simple, inference-only algorithm that operates efficiently with just 30 random samples.
- These selected norms are then used in a straightforward voting algorithm, enhancing the model's prediction accuracy by treating head norms as an ensemble of weak learners.

NoVo's approach avoids reliance on specialized tools or in-domain training, making it scalable and generalizable. The method achieves state-of-the-art performance on the TruthfulQA MC1 benchmark, surpassing previous methods by at least 19 accuracy points. Additionally, NoVo demonstrates strong generalization across 20 diverse datasets, significantly outperforming existing representation editing and reading techniques, and showcasing its robustness in improving LLM factual accuracy.

**Strengths:**

- NoVo is designed to be lightweight and does not require specialized tools, in-domain training, or external resources. This simplicity allows it to scale effortlessly across diverse datasets and applications, making it practical for real-world deployment.

- The method achieves remarkable accuracy gains, notably setting a new state-of-the-art on the TruthfulQA MC1 benchmark with a 19-point improvement over existing approaches. This demonstrates the effectiveness of NoVo in addressing the critical issue of hallucinations in LLMs.

- NoVo exhibits exceptional generalizability, achieving significant accuracy improvements on over 90% of the 20 diverse datasets evaluated. This indicates the method’s robustness and versatility in handling a wide range of tasks beyond just one specific dataset or domain.

**Weaknesses:**

- While the paper presents detailed analysis and shows impressive gains across benchmarks, the applicability of the solution beyond the MCQ type of problems is not obvious.
- The motivation to solve MCQ type questions is not clear. Why is it very important to solve?

**Questions:**

Authors claim "Hallucinations in Large Language Models (LLMs) remain a major obstacle, particularly in high-stakes applications where factual accuracy is critical". Are the benchmarks used in this study representative of the same?

We see models being adopted to simple applications that involve RAG, QnA etc. or more complex Agentic use cases that involve abilities like function calling, planning etc.

It is not very clear how the benchmarks help in high-stakes applications and it i very difficult assess the impact of the work.

Another question that arises is - how critical is MCQ based tasks? should we even solve it?

---

> ### Author Response · Authors · 2024-11-20
> **Official Response to Reviewer 3eKd (1/4)**
>
> **Q1**: While the paper presents detailed analysis and shows impressive gains across benchmarks, the applicability of the solution beyond the MCQ type of problems is not obvious.
>
> **A1**: MCQ tasks are not a limitation of our method but rather are used as an evaluation tool to explore and confirm any fundamental internal state misalignments. There are four applications of our solution beyond MCQ tasks:
> 1. **Ranked Generation** Our method can be applied beyond MCQs to other task types, such as ranking multiple candidate text spans either generated, or retrieved, by the LLM. Our analysis reveals that head norms spike significantly by up to 83% when compared to a different candidate text, in token positions where the proposition is complete (Lines 516-527, 389-402, 430-435, 954-1011). This phenomenon can be used to rank the truthfulness of candidate text spans during open-ended generation.
>
> 2. **Hallucination Detection** Our findings can be applied beyond MCQs to showcase a more fundamental problem: common and fluent falsehoods are strongly indicative of internal state misalignments. TruthfulQA is a benchmark specifically crafted to mislead LLMs to output common and fluent falsehoods. We show an average 24 point accuracy gain across eight models$^{1}$ in TruthfulQA with head norms, over the language likelihood (Lines 291-298). This difference can be used in hallucination detection tasks, especially for factually incorrect but fluent outputs.
>
> 3. **Adversarial Defence** Our method can be applied beyond MCQs to improve model robustness against textual attacks. An average 6.6 median accuracy gain across four models on all six subtasks of AdversarialGLUE showcase the potential of head norms for building textual adversarial defence (Lines 279-282, 287, 317-323).
>
> 4. **Finetuning** Our method can be applied to improve finetuning accuracy on general tasks beyond MCQs, such as during the alignment stage for better open-ended generations. Experiments on head-norm-finetuning with DeBERTa show an average 0.9 point accuracy gain over standard feature vector fine-tuning, across all nine datasets.
> &nbsp;
>
> $^{1}$ The evalutions of four new models have been added, doubling the number of models from the initial four to eight.

---

> ### Author Response · Authors · 2024-11-20
> **Official Response to Reviewer 3eKd (2/4)**
>
> **Q2**: The motivation to solve MCQ type questions is not clear. Why is it very important to solve?
> **A2**: There are five reasons why solving MCQ type questions is important and critical.
>
> 1. **A Fundamental Task** According to Bloom’s Taxonomy of cognitive pedagogy [1], recognising and understanding the correct answer are the two most fundamental learning outcomes. These are present in MCQ tasks. To analyse, apply and synthesise are higher-level outcomes which are present in open generation. Fundamental MCQ benchmarks like TruthfulQA remain unsolved even with models like GPT4. Furthermore, there is scrutiny in a LLM’s true ability to solve generative tasks [2]. Before moving up to more advanced outcomes, It is critical to solve MCQ tasks to determine an LLM’s fundamental capabilities.
>
> 2. **Real-life Applicability** MCQ tasks are used in the real high-stakes applications as a key component of standardised assessments for professional certifications and regulatory compliance [3-6]. These MCQs require specific formatting to ensure clarity and fair comparability, whereas open-ended generation could introduce interpretative ambiguity. Solving MCQ tasks in LLMs is an important step towards their use in real-world, high-stakes scenarios, where standardised testing is necessary for regulatory compliance and trustworthiness.
>
> 3. **Academic Significance** MCQ tasks allow us to objectively evaluate the factual knowledge, reasoning, and understanding skills of an LLM with a direct and unified metric (accuracy). By choosing MCQ tasks as an evaluation tool, we are able to assess our observations of internal state misalignment across significantly more varied benchmarks, models, and methods, without needing any nuisance control over prompting and evaluation techniques. It is important to solve these MCQ benchmarks to directly and objectively validate our approach versus the rest.
>
> 4. **Technically Challenging** MCQ tasks are equally challenging to solve when compared to generative tasks. Well-designed MCQs can include answer distractors, ambiguity, and misleading phrasing to confuse LLMs. The constrained output space of MCQ tasks might seem easier when compared to generating novel responses, but performance on benchmarks do not reflect this. Measuring letter option probability in the logit space is prone to selection bias [7], while finetuning on MCQ examples causes overfitting and forgetting [8]. The canonical way is to select the answer option with the highest language likelihood [9], but accuracy is still far from human performance in some benchmarks like TruthfulQA [10]. It is important to solve these technical challenges posed by MCQs to better understand and overcome these fundamental limitations of LLMs.
>
> 5. **Benefits for Generation** Success on MCQ tasks reflects an LLM's ability to recognise coherent, reasonable and factual statements, under challenging questions. These abilities form the foundation for open-ended generation, whereby similar capabilities are required but without the explicit constraints of predefined options. Solving MCQ tasks thus serves as a stepping stone and evaluation method for enhancing performance in more complex generative scenarios.
> &nbsp;
> ##### [1] Bloom, B. S., Engelhart, M. D., Furst, E. J., Hill, W. H., & Krathwohl, D. R. (1956). Taxonomy of Educational Objectives: The Classification of Educational Goals, Handbook I: Cognitive Domain.
> ##### [2] GSM-Symbolic: Understanding the Limitations of Mathematical Reasoning in Large Language Models, Arxiv 2024.
> ##### [3] Occupational Safety and Health Administration, Hazard Identification and Assessment, accessed online at https://www.osha.gov/safety-management/hazard-identification.
> ##### [4] Med-HALT: Medical Domain Hallucination Test for Large Language Models, EMNLP 2023.
> ##### [5] Multistate Bar Examination, National Conference of Bar Examiners, accessed online at https://www.ncbex.org/exams/mbe.
> ##### [6] Graduate Record Examinations, Educational Testing Service, accessed online at https://www.ets.org/gre.html.
> ##### [7] Large Language Models Are Not Robust Multiple Choice Selectors, ICLR 2024.
> ##### [8] An Empirical Study of Catastrophic Forgetting in Large Language Models During Continual Fine-tuning, Arxiv 2024.
> ##### [9] Language Models are Unsupervised Multitask Learners, OpenAI Blog 2019.
> ##### [10] TruthfulQA MC1 https://paperswithcode.com/sota/question-answering-on-truthfulqa.

---

> ### Author Response · Authors · 2024-11-20
> **Official Response to Reviewer 3eKd (3/4)**
>
> **Q3**: Authors claim "Hallucinations in Large Language Models (LLMs) remain a major obstacle, particularly in high-stakes applications where factual accuracy is critical". Are the benchmarks used in this study representative of the same?
> **A3**: Yes, our diverse range of benchmarks is representative of the different facets of factual hallucinations, a critical issue in high-stakes applications [1-3]. Our method was designed to be generalisable and easily applied to a wide range of specialised datasets, such as those designed for strategic reasoning in multi-turn dialogues (Line 257), textual adversarial attacks (Line 259), domain-specific knowledge testing (Lines 255, 260-263, and causal reasoning over narrative contexts (Line 256). The benchmarks used in this study show that our method enables LLMs to elicit the correct factual knowledge under adversarial or misleading contexts, which is crucial for high-stakes applications.
> &nbsp;
> [1] Capabilities of GPT-4 on Medical Challenge Problems, Arxiv 2023.
> [2] Performance of ChatGPT on USMLE: Potential for AI-assisted medical education using large language models. PLoS Digital Health 2, 2023.
> [3] Large language models encode clinical knowledge, Nature 2023.

---

> ### Author Response · Authors · 2024-11-20
> **Official Response to Reviewer 3eKd (4/4)**
>
> **Q4**: It is not very clear how the benchmarks help in high-stakes applications and it is very difficult to assess the impact of the work.
>
> **A4**: Our contributions go beyond specific benchmark performances to showcase and tackle fundamental issues that can arise in high-stake scenarios. The impact of our work with these benchmarks can be summarised in four points
>
> 1. **Using Benchmarks to Demonstrate Internal Misalignments** Our results on various benchmarks show that internal misalignments contribute to factual hallucination across a wide and diverse range of topics. We also show that the log likelihood tends to favour fluent but incorrect output, and that this limitation can be addressed with head norms. Our benchmark results indicate that this undesirable phenomenon can arise even in high-stake scenarios, where the cost of hallucinations is much higher.
>
> 2. **Tackling Inherent Weaknesses in LLMs on an unsolved benchmark** TruthfulQA MC1 remains unsolved at 60% even with GPT4, due to its misleading questions. Our method showcases that this limitation of LLMs can be addressed with internal states (head norms). Accuracy gains on this benchmark is not only academically significant but also serves as a critical solution in high-stakes scenarios, where misleading questions can have expensive and harmful consequences.
>
> 3. **Applicability to High-Stakes Testing** In high-stakes applications, it is crucial for LLMs to undergo standardised testing for regulatory compliance as part of the trustworthiness framework [1]. These tests often contain MCQ questions to ensure clear and fair evaluations [2-5]. Our results on a wide variety of benchmarks showcase the generalizability of our method on different MCQ domains and formatting styles, which is useful when translating to high-stakes, real-world unconstrained MCQ styles.
>
> 4. **Potential for Adversarial Defence** We report an average 6.6 median accuracy gain across four models on all six subtasks of AdversarialGLUE. These results indicate the potential for head norms to be used to enhance textual attack robustness. This benchmark is particularly relevant in high-stakes scenarios, where bad-faith actors can inject adversarial prompts to elicit undesirable outputs.
> ---
> ---
> **Q5**: Another question that arises is - how critical is MCQ based tasks? should we even solve it?
>
> **A5**: MCQ based tasks are critical to solve, because they are:
> - Fundamental tasks that reflect low-level learning outcomes.
> - Applicable in real-life standardised testing for regulatory compliance of LLMs.
> - Academically significant in objective comparison between different approaches.
> - Technically challenging tasks that probe at a LLM’s limitations.
> - Offer foundational benefits for open-ended generation.
>
> We gently refer the reviewer to the more detailed Answer 2 (A2) to Question 2 (Q2).
>
> &nbsp;
> [1] Biswas, A., & Talukdar, W. (2023). Guardrails for trust, safety, and ethical development and deployment of Large Language Models (LLM). Journal of Science & Technology, 4(6), 55-82.
> [2] Occupational Safety and Health Administration, Hazard Identification and Assessment, accessed online at https://www.osha.gov/safety-management/hazard-identification.
> [3] Med-HALT: Medical Domain Hallucination Test for Large Language Models, EMNLP 2023.
> [4] Multistate Bar Examination, National Conference of Bar Examiners, accessed online at https://www.ncbex.org/exams/mbe.
> [5] Graduate Record Examinations, Educational Testing Service, accessed online at https://www.ets.org/gre.html.

---

> ### Author Response · Authors · 2024-11-25
> **Gentle Reminder: Period Ending – We Anticipate Your Feedback!**
>
> Dear Reviewer 3eKd,
>
> Thank you again for your comments, effort and time spent reviewing this paper.
>
> We gently remind the reviewer that the discussion stage is ending. We deeply appreciate your valuable insights in helping us improve our manuscript's quality. Once this stage ends, we will be unable to respond.
>
> Kindly let us know if you have further comments about this paper. We look forward to your suggestions!
>
> Regards,
> Authors of Submission 3392

---

> ### Author Response · Authors · 2024-11-29
> **Happy to Engage in Further Discussions**
>
> Dear Reviewer 3eKd,
>
> We understand you might be busy! We would be deeply grateful if you could kindly take a moment to share your thoughts. Your acknowledgment and further feedback would mean a great deal to us.
>
> - We summarise our responses and paper revisions to address all your four (4) concerns:
>
> **1. Applicability Beyond MCQs?**
> ```Response``` Our solution can potentially be applied beyond MCQ tasks to ranked generation, hallucination detection, adversarial defence and finetuning for alignment.
> ```Revision``` Detailed how our solution can be applied beyond MCQ tasks in Sections 1 and 4.4.
>
> **2 Why Solve MCQs?**
> ```Response``` Solving MCQs is crucial because they probe fundamental and low-level learning outcomes, are commonly used in high-stakes-scenario standardised assessments for regulatory compliance, provides a unified and objective metric across a wide range of benchmarks for fair comparison against other methods, are technically challenging tasks that reveal fundamental limitations of models, and forms the foundational towards more complex generative scenarios.
> ```Revision``` Detailed our motivation to solve MCQ tasks and explained why MCQs are critical in Section 1.
>
> **3 Do your Benchmarks Represent High-Stake Scenarios?**
> ```Response``` Yes, our diverse range of benchmarks comprehensively probes different facets of factual hallucinations, a critical issue in high-stakes scenarios. This includes strategic reasoning in multi-turn dialogues, textual adversarial attacks, general and scientific knowledge, causal reasoning over narratives, social and physicality reasoning, and responding under misleading contexts. Good performance in these benchmarks reflect crucial skills for high-stakes scenarios.
> ```Revision``` Explicitly showed how our benchmark performance are representative of high-stakes applications in Section 1.
>
> **4 Impact of Your Work?**
> ```Response``` We demonstrate a fundamental problem: internal state misalignments with the language likelihood, which impacts future works in deeply addressing hallucinations. We achieve a significant 30-point accuracy gain in an unsolved benchmark and generalize to 20 more datasets, which impacts future efforts in devising more accurate and generalizable methods. Our benchmark results can impact real-world high-stakes scenarios, where models must undergo standardised MCQ testing for regulatory compliance as part of the trustworthiness framework. Our results on AdversarialGLUE and finetuning may impact future works in adapting our method for adversarial defence and safety-alignment finetuning.
> ```Revision``` Added an impact statement which contextualized our benchmark performances to real-world high-stake scenarios in Section 1.
> $~$
> We’d be delighted to receive any further thoughts and perspective from you.
> Thank you so much for your time and consideration.
>
> _Authors of Submission 3392_

---

### Author Response · Authors · 2024-11-21
**General Response: Revision Summary**

We sincerely thank all reviewers for their valuable and constructive suggestions in improving our paper. Based on all reviewer suggestions, we are happy to upload a revised pdf of our paper. Changes are highlighted in **Blue**.

Summary of revisions:

- Suggestions from Reviewer **3eKd**.
    1. Added four applications of our solution beyond MCQ tasks in Section 1 and Section 4.4.
    2. Added explanation of motivation to solve MCQ tasks in Section 1.
    3. Explicitly showed how our benchmark performances are representative of high-stakes applications in Section 1.
    4. Translated our benchmark performances to an impact statement in high-stake scenarios in Section 1.
    5. Included paragraph explaining why MCQ based tasks are critical in Section 1.

- Suggestions from Reviewer **GDLP**.
    1. Revised Section 3.1 to better explain why the correlation between head norms and truth was reasonably expected.
    2. Added four applications of our solution beyond MCQ tasks in Section 1 and Section 4.4.
    3. Added new Appendix I to evaluate four new models.
    4. Revised Section 4.3 to better explain why results vary across datasets.
    5. Added a new paragraph of random variations in existing Appendix D.

- Suggestions from Reviewer **a4Cj**.
    1. Added four applications of our solution beyond MCQ tasks in Section 1 and Section 4.4.
    2. Added new Appendix H to evaluate norms on MLP and other hidden states.
    3. Revised last paragraph of Section 5.1 to better explain why such a correlation could exist.
    4. Revised Section 3.1 to better explain why the correlation between head norms and truth was reasonably expected.
    5. Added new paragraph about OOD performance in existing Appendix B.
    6. Revised last paragraph of Section 1 to better align our summarised contributions with interpreting and improving multi-choice scenarios.

---

### Author Response · Authors · 2024-11-22
**Invitation to Reviewer-Author Discussion**

Dear Reviewers,

We sincerely thank all of you for your valuable and insightful comments. We greatly appreciate your constructive feedback in improving our paper.

We invite all reviewers for further discussions of our paper and are happy to respond to any further questions. Your active participation will be crucial in ensuring well-informed decisions.

Once again, we thank you all for your time and effort,

Authors of Submission 3392

---

### Meta-Review · Area_Chair_7PkG · 2024-12-21

**Metareview:**

The paper introduces NoVo, a lightweight method for reducing hallucinations in LLMs using attention head norms. By leveraging these norms with a voting mechanism, NoVo achieves significant performance improvements on multiple-choice QA tasks and generalizes well across diverse datasets. Reviewers praised the simplicity, scalability, and robust empirical results. I recommend acceptance.

**Additional Comments On Reviewer Discussion:**

Reviewers' concerns were raised regarding the theoretical justification of attention head norms, applicability to open-ended tasks, and variability in results. The authors addressed these issues with additional experiments and explanations. The reviewer 3eKd's concerns are a bit biased and have already been addressed during the response.

---

### Decision · Program_Chairs · 2025-01-22

Accept (Poster)